# Mechanisms of virus dissemination in bone marrow of HIV-1–infected humanized BLT mice

Mark S Ladinsky[1†], Wannisa Khamaikawin[2†‡], Yujin Jung[2§], Samantha Lin[2], Jennifer Lam[2#], Dong Sung An[2], Pamela J Bjorkman[1], Collin Kieffer[1¶]*

[1]Division of Biology and Biological Engineering, California Institute of Technology, Pasadena, United States; [2]School of Nursing, UCLA AIDS Institute, University of California, Los Angeles, Los Angeles, United States

**Abstract** Immune progenitor cells differentiate in bone marrow (BM) and then migrate to tissues. HIV-1 infects multiple BM cell types, but virus dissemination within BM has been poorly understood. We used light microscopy and electron tomography to elucidate mechanisms of HIV-1 dissemination within BM of HIV-1–infected BM/liver/thymus (BLT) mice. Tissue clearing combined with confocal and light sheet fluorescence microscopy revealed distinct populations of HIV-1 p24-producing cells in BM early after infection, and quantification of these populations identified macrophages as the principal subset of virus-producing cells in BM over time. Electron tomography demonstrated three modes of HIV-1 dissemination in BM: (*i*) semi-synchronous budding from T-cell and macrophage membranes, (*ii*) mature virus association with virus-producing T-cell uropods contacting putative target cells, and (*iii*) macrophages engulfing HIV-1–producing T-cells and producing virus within enclosed intracellular compartments that fused to invaginations with access to the extracellular space. These results illustrate mechanisms by which the specialized environment of the BM can promote virus spread locally and to distant lymphoid tissues.
DOI: https://doi.org/10.7554/eLife.46916.001

*For correspondence:
collink@illinois.edu

†These authors contributed equally to this work

Present address: ‡Faculty of Medicine, King Mongkut's Institute of Technology Ladkrabang, Bangkok, Thailand; §Department of Biomedical Sciences, Cedars Sinai Medical Center, Los Angeles, United States; #Johns Hopkins University School of Nursing, Baltimore, United states; ¶Department of Microbiology, University of Illinois at Urbana-Champaign, Urbana, United States

## Introduction

The rapid systemic dissemination of HIV-1 from the site of initial infection to distant lymphoid tissues is a critical component of the acute phase of HIV-1 infection. Bone marrow (BM) within the spongy matrix of long bones functions as both a primary and secondary lymphoid organ and is the main site for hematopoiesis of immune cells destined for tissues throughout the body. It contains cells commonly associated with HIV-1 pathogenesis (e.g., CD4+ T-cells, macrophages, and dendritic cells) and BM-specific cells such as megakaryocytes and committed lymphoid and myeloid progenitor cells that are also permissive to HIV-1 infection (*Alexaki and Wigdahl, 2008*; *Allen and Dexter, 1984*; *Folks et al., 1988*). HIV-1 infection leads to abnormalities in hematopoiesis (*Redd et al., 2007*; *Thiebot et al., 2001*; *Yamakami et al., 2004*), and virus replication can be detected in the BM of SIV-infected non-human primates (NHPs) within days after infection (*Mandell et al., 1995*). Hematopoietic stem cells (HSCs) in BM can differentiate into multiple HIV-1 permissive cell types that retain the capacity to enter the bloodstream and travel to distant tissue sites in a controlled fashion (*Geissmann et al., 2010*; *Pabst, 2018*; *Pittet et al., 2014*), a process that could facilitate systemic virus dissemination. The presence of varied BM cell types permissive to HIV-1 infection, the ability of BM cells to migrate systemically, and the proposed implication of HSCs and progenitor cells (HSPCs) as cell types that contribute to the virus reservoir (*Carter et al., 2011*; *Carter et al., 2010*; *McNamara et al., 2013*), make BM an important, but poorly understood, tissue for probing biological mechanisms of HIV-1 dissemination and pathogenesis.

 

Humanized mice (hu-mice) are a cost-effective model for imaging HIV-1 within tissues and the only HIV-1 infection from which one can routinely obtain BM from long bone samples. Studies in HIV-1–infected hu-mice have recapitulated and advanced important observations of HIV-1 disease, including acute spread and systemic dissemination, latency, and target cell depletion (*Marsden and Zack, 2015*; *Marsden and Zack, 2017*). Several hu-mouse models have been used to study HIV-1 pathogenesis, with BM, liver, thymus (BLT) hu-mice arguably containing the most complete repertoire of human immune cells of hu-mouse models (*Denton and Garcia, 2012*; *Marsden and Zack, 2015*). BLT mice are individually created by transplanting human liver and thymus tissues together with autologous CD34+ HSCs into NOD/SCID/IL2Rγ null (NSG) mice. Importantly, BM from HIV-1–infected BLT hu-mice includes human immune cells that produce virus transcripts during infection (*Denton et al., 2008*; *Nixon et al., 2013*).

Immunofluorescence (IF) microscopy can be used to survey specific cell populations within tissues of BLT hu-mice and other models of HIV-1 infection. Larger volume imaging (mm$^3$-cm$^3$) is possible using tissue clearing methods, which render samples effectively transparent by removing lipids and other biomolecules to enhance light penetration into the tissue (*Richardson and Lichtman, 2015*; *Tainaka et al., 2016*; *Treweek and Gradinaru, 2016*). Combining tissue clearing with light sheet fluorescence microscopy (LSFM) allows in situ interrogation of large tissue volumes with single cell resolution and rapid generation of quantifiable spatial information that would be difficult to achieve with traditional immunohistochemistry techniques (*Richardson and Lichtman, 2015*; *Tainaka et al., 2014*; *Yang et al., 2014*). Tissue clearing has allowed imaging of brain connectivity in mice (*Chung et al., 2013*; *Susaki et al., 2014*), whole-body metastasis in mouse models of cancer (*Guldner et al., 2016*; *Kubota et al., 2017*), and the distribution of HIV-1–infected cells within gut-associated lymphoid tissue (GALT) and spleen (*Kieffer et al., 2017a*; *Kieffer et al., 2017b*). Although the inherent density of bone prevents adequate clarification using many tissue clearing techniques, BM imaging is possible using Bone CLARITY, which was developed to clear mouse femurs in order to address the spatial distribution of specific cell populations (*Greenbaum et al., 2017*).

Lymphoid tissues can be imaged at higher resolution using electron microscopy (EM), with electron tomography (ET) allowing 3D visualization of tissue volumes at ultrastructural resolution (*McIntosh et al., 2005*). We previously reported ET studies of HIV-1 dissemination in hu-mouse GALT (*Ladinsky et al., 2014*) and a longitudinal study of HIV-1 in hu-mouse GALT and spleen during acute infection (*Kieffer et al., 2017b*). After using immuno-EM to verify the identity of virions in tissue as HIV-1, we used ET of optimally-preserved tissues to distinguish, characterize and quantify free mature virions, free immature virions, and budding virions within 3D tissue volumes (*Kieffer et al., 2017b*; *Ladinsky et al., 2014*). These studies revealed 3D profiles of free and budding HIV-1 virions and documented large pools of extracellular cell-free virions localized between cells and occasional instances of virological synapses, supporting roles for both cell-free and cell-to-cell virus dissemination in these hu-mouse tissues (*Kieffer et al., 2017b*; *Ladinsky et al., 2014*). In contrast, our ET studies of murine leukemia virus (MLV) in secondary lymphoid mouse tissues showed extensive zones of contact between infected macrophages and B-cells, virions in the cell-cell interface (virological synapses), and transfer of virions from infected donor cells to uninfected target cells via uropod extensions (*Sewald et al., 2015*). The potential roles of cell-free versus cell-associated virus transmission via cellular structures for HIV-1 in BM remain unexplored.

To address mechanisms of HIV-1 dissemination in BM, we employed a multiscale imaging approach: Bone CLARITY (*Greenbaum et al., 2017*) combined with IF, confocal, and LSFM to quantify the distributions of specific cell types associated with HIV-1 within intact volumes of BM; and ET to detect individual virions and virion-producing cells to characterize aspects of HIV-1 infection at the subcellular level. These imaging studies allowed visualization of HIV-1 distribution within larger volumes of tissue from a hu-mouse model and revealed distinct mechanisms of virus dissemination within BM.

## Results

### Generation of HIV-1–infected BLT mice

To investigate mechanisms of HIV-1 dissemination in BM, BLT hu-mice were generated by transplantation of human fetal liver-derived CD34+ HSPCs via retro-orbital vein plexus injection and transplantation of pieces of human liver and thymus under the mouse kidney capsule in 8-week-old irradiated NSG mice. Human multi-lineage hematopoietic cell populations (hCD45+ hematopoietic, hCD3+ T-cell, hCD19+ B-cell, hCD4+ T-cell and hCD8+ T-cell) were reconstituted in peripheral blood at 13 weeks post transplantation (*Figure 1—figure supplement 1A,B*). BLT hu-mice were infected with CCR5-tropic HIV-1$_{NFNSX}$ at a dose of 200 ng of p24 gag via retro-orbital vein plexus at 14 weeks. Animals were euthanized and sternums and femurs were harvested to investigate HIV-1 dissemination in BM at 5 days (n = 1), 10 days (n = 2), 63 days (n = 1) and 92 days (n = 1) post-infection (*Figure 1—figure supplement 1C*). The hCD4/CD8 ratios in peripheral blood were unaltered at 5 and 10 days post-infection and subsequently declined to low levels (*Figure 1—figure supplement 1D*). Accordingly, the HIV-1 viral load in peripheral blood was low at 0 (m410) and 5 (m419) days post-infection, but rapidly reached greater than $1.25 \times 10^5$ copies of HIV-1 RNA/mL for all later timepoints (*Figure 1—figure supplement 1E*).

### IF surveys of BM cell distributions from HIV-1-infected hu-mice

Sternums and femurs from HIV-1–infected BLT hu-mice were fixed and cleared by Bone CLARITY (*Greenbaum et al., 2017*) (*Figure 1A*).~1–2 mm regions of cleared sternum containing a central channel of BM were imaged by confocal fluorescence microscopy after immunostaining for cell nuclei, HIV-1 p24, human CD4 (hCD4)-expressing cells, and human CD68 (hCD68)-expressing macrophages (*Figure 1B,C*). Compared with other lymphoid tissues, BM does not contain a dense distribution of cells (*Figure 1B*). This approach allowed us to localize hCD4+ cells, hCD68+ macrophages, and cells that were negative for both markers in addition to highlighting the stark differences in size and morphology exhibited by these cell types. (*Figure 1C*). HIV-1 p24+ co-localized with both hCD4+ and hCD68+ cells. These results showed that hCD4+ and hCD68+ cells are present within regions of intact, cleared BM from HIV-infected BLT hu-mice, and confirmed HIV-1 p24 protein expression associated with both cell types as evidenced by co-staining with HIV-1 p24.

To obtain information from larger BM volumes, we analyzed femurs from HIV-1–infected hu-mice. Low-level autofluorescence from cleared and decolorized femurs was recorded using LSFM to generate a reference volume for imaging after immunostaining (*Figure 1D*). Imaging of a ~ 5 mm long and ~2 mm diameter region of immunostained femur from a 10 day post-infection (PI) hu-mouse showed numerous hCD4+ and hCD68+ cells dispersed throughout the BM but relatively few p24+ cells (*Figure 1D*). Segmentation and quantification to detect the density of the individual cell populations showed that 626 per mm$^3$ hCD4+ cells, 766 per mm$^3$ hCD68+ T-cells, and five per mm$^3$ p24+ cells were present within the sample volume. Additional timepoints PI were imaged and quantified, demonstrating a constant density of hCD68+ cells, a decreased density of hCD4+ cells, and a low density of p24+ cells that increased slightly over time (*Figure 1E*). The reduction of hCD4+ T-cells is consistent with NHP studies showing SIV-induced T-cell depletion in BM (*Hoang et al., 2019*). Co-localization analysis indicated that the majority of p24+ cells were hCD68+, with this level remaining consistent over time, whereas the percentage of p24+/hCD4+ cells decreased with time (*Figure 1E*).

### ET surveys of cell and virus distributions in infected BM

In direct comparisons of ET imaging of BM obtained by needle aspiration versus after dissection and removal from long bones, we found that BM extracted by needle aspiration was contaminated with blood from the vasculature, and we could only obtain physiologically-relevant regions of BM that were distinct from contaminating mature red blood cells at the EM level by extracting BM samples from long bones (unpublished results). We therefore used intact BM tissue taken from HIV-1–infected BLT mice after dissection and removal from long bones for ET. Specifically, BM tissues were obtained by removing ~1 cm of the sternum bone immediately after the mouse was sacrificed and then placing it in a fixative after which the sternum was opened and intact BM was removed without disruption of the underlying tissue structure.

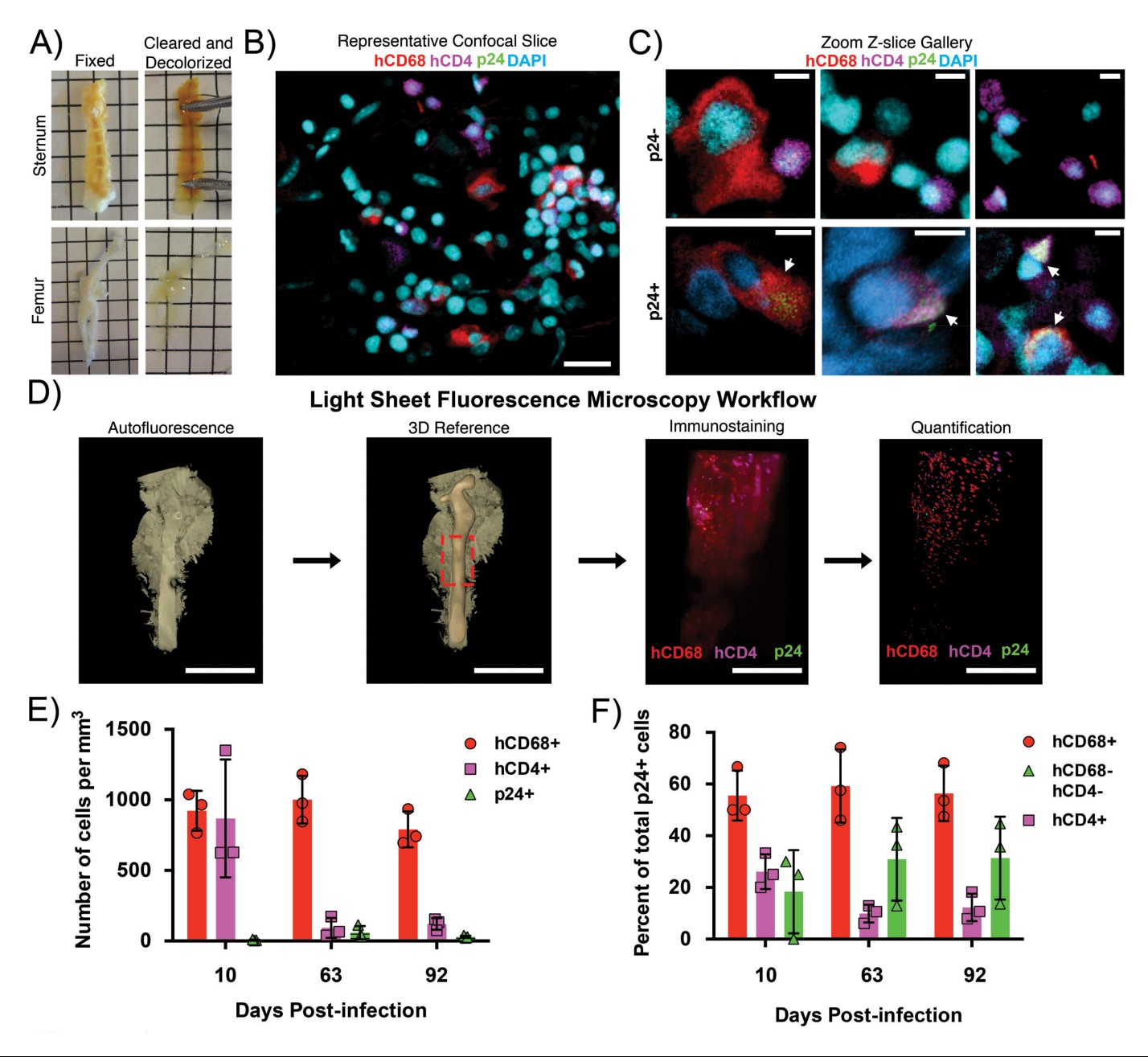

**Figure 1.** Clearing and imaging of HIV-1–infected BLT BM. (**A**) Examples of fixed (left) and cleared/decolorized (right) sternums and femurs from HIV-1-infected BLT hu-mice. Squares are 5 mm by 5 mm. Images of tissues before and after clearing were captured on an iPhone 5 (Apple). (**B**) Representative confocal Z-slice of cleared sternum from an HIV-1–infected hu-mouse 63 days post-infection immunostained for hCD4 (magenta), hCD68 (red), HIV-1 p24 (green), and nuclei (blue). (**C**) Gallery of zoomed confocal Z-slices showing characteristic sizes and morphologies of hCD4+ cells (magenta), hCD68 + cells (red), and p24+ cells (green). Arrows indicate p24+ cells. (**D**) Volume of autofluorescence from cleared femur of an HIV-1 infected hu-mouse 10 days post-infection captured with light sheet microscope (left). Segmented model of bone from the same volume (center left). Dashed red box shows region of interest for subsequent panels. Light sheet volume of a region of BM immunostained for hCD4 (magenta), hCD68 (red), and HIV-1 p24 (green) (center right). Segmented model from the same dataset showing individual cell distributions within BM (right). (**E**) Quantification of individual cell population densities over time. (**F**) Percent of total p24+ cells co-localizing with hCD68+ cells (red), hCD4+ cells (magenta), or not co-localized with hCD68+ or hCD4+ cells (green) at specific times post-infection. Error bars represent standard deviations from the mean of 3 measurements from separate volumes of tissue greater than 0.5 mm³ each. Scale bars: (B = 20 µm; C = 5 µm; D = 4 mm, left panels; 500 µm, right panels).

DOI: https://doi.org/10.7554/eLife.46916.002

The following figure supplement is available for figure 1:

*Figure 1 continued on next page*

*Figure 1 continued*

**Figure supplement 1.** Human cell reconstitution and HIV-1 infection in BLT hu-mice.

DOI: https://doi.org/10.7554/eLife.46916.003

BM from sternums of HIV-1–infected BLT hu-mice was prepared for ET by prefixation with glutaraldehyde and paraformaldehyde followed by high-pressure freezing and freeze-substitution (HPF/FS) and resin embedding (*Kieffer et al., 2017b*; *Ladinsky et al., 2014*). Although we cannot use the same samples for high-resolution ET and immuno-EM because antibody epitopes on resin-embedded samples are typically destroyed (*Ladinsky and Howell, 2007*), many of the cells in BM can be identified by morphology as we describe below. With the exception of not including budding or free virions, uninfected BLT mouse BM was not distinguishable from HIV-1–infected BLT mouse BM in terms of cell density, cell types, or cell morphologies (data not shown).

Large areas of HIV-1–infected BM were surveyed as EM overviews comprising 50–500 montaged frames at ~6500 x magnification (*Figure 2A*). These surveys revealed a diverse array of cell morphologies consistent with cells found in BM, including T-cells, B-cells, macrophages, megakaryocytes, and HSCs/HSPCs (*Figure 2A*; *Figure 2—figure supplement 1*). Compared with tissues such as

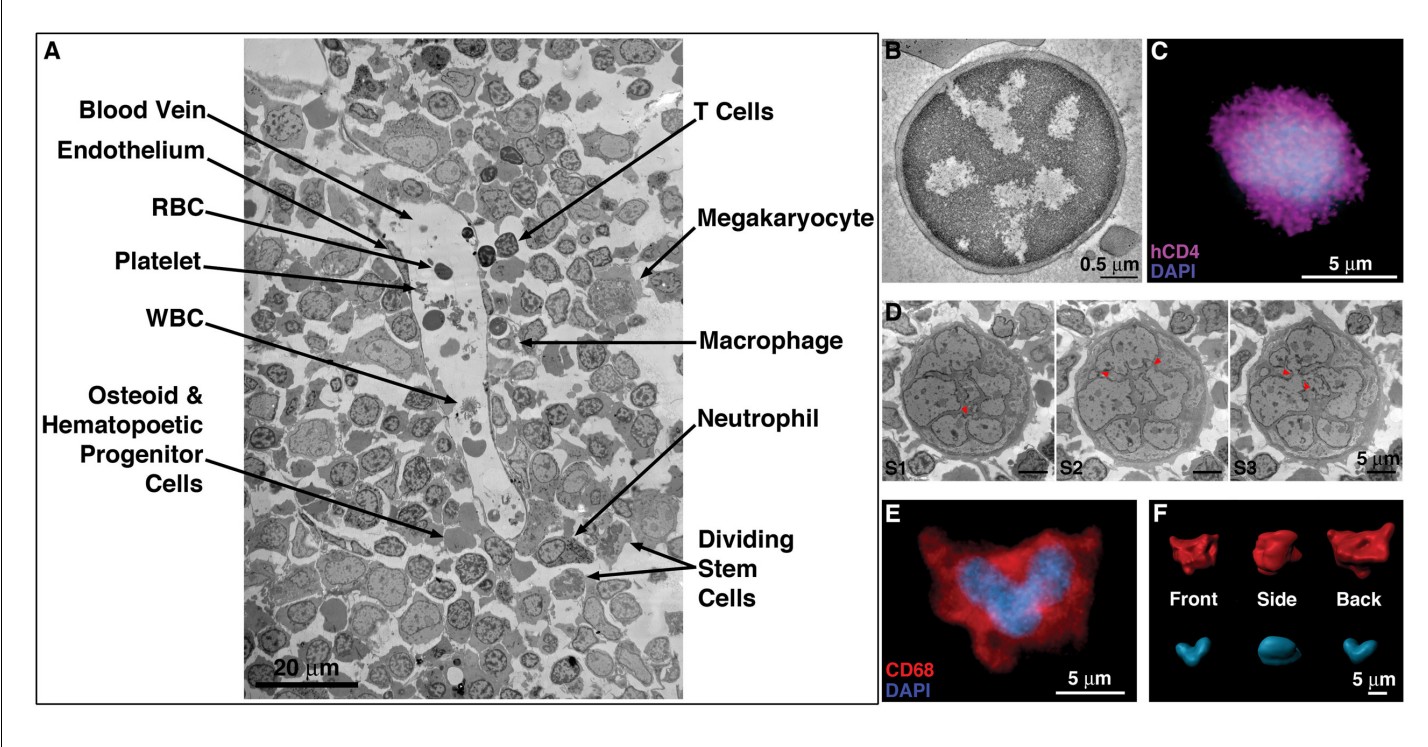

**Figure 2.** EM and LM imaging of cell types found in BM. (A) EM overview of a vascularized region, showing a variety of resident cell types within BM (labeled). (B) EM tomographic slice showing typical morphology of a BM T-cell (spherical shape and large nucleus-to-cytoplasm ratio). (C) Confocal IF image showing an hCD4+ cell with a spherical shape and large nucleus-to-cytoplasm ratio. (D) Montaged projection EM overviews of a polymorphonuclear BM macrophage in three serial 400 nm sections, demonstrating connectivity between the various lobes of the complex nucleus (arrowheads). (E) Representative confocal slice of an hCD68+ cell (red) showing a multilobed nucleus (blue). Distinctly multinuclear macrophages were not observed in BM. (F) Segmented 3-D volume of the cell in (E) showing the complex surface morphology (red) and a single nucleus (blue) with multiple interconnected lobes.

DOI: https://doi.org/10.7554/eLife.46916.004

The following figure supplements are available for figure 2:

**Figure supplement 1.** T-cells and macrophages display different modes of cell-to-cell contact for potential virus transfer.

DOI: https://doi.org/10.7554/eLife.46916.005

**Figure supplement 2.** HIV-1–bearing uropods are associated with macrophages.

DOI: https://doi.org/10.7554/eLife.46916.006

GALT and spleen, the density of cells in hu-mouse BM was relatively low, as also found for human BM (*Brass, 2005*). IF of an equivalent region of hu-mouse BM showed a similar distribution of cells, including hCD4+ cells and hCD68+ cells, dispersed amongst other cell types (*Figure 1B*). By EM, T-cells can be distinguished by their round compact shape, large nucleus, and minimal cytoplasm (*Zucker-Franklin, 1975*) (*Figure 2B*). Cells identified by EM as T-cells were morphologically similar to hCD4+ cells detected by IF within cleared BM (*Figure 1C*; *Figure 2C*). Cells identified as BM T-cells regularly displayed uropods (cytoplasmic protrusions containing organelles and structures promoting cell adhesion and signaling; *Sánchez-Madrid and Serrador, 2009*; *Sewald et al., 2015*) that formed synaptic contacts with neighboring cells (*Figure 2—figure supplement 1A,G*), suggestive of the migration characteristics exhibited by some innate immune cells (*Hind et al., 2016*; *Lämmermann and Sixt, 2009*; *Sewald et al., 2015*).

Macrophages are larger than T-cells and are characterized by a complex morphology that includes deep surface invaginations, multi-lobed nuclei, and populations of vesicles and granules of varying size and electron density (*Orenstein, 2007*; *Orenstein and Wahl, 1999*) (*Figure 1C*; 2D-F). Although some IF and projection EM images appeared to show multiple nuclei in presumptive macrophages, serial-section ET of large 3D volumes demonstrated that individual lobes of nuclei were connected in the ~25 cells that we examined (*Figure 2D*), suggesting that BM macrophages are not polynuclear; that is they do not contain multiple separate nuclei. Analogous IF data from cleared BM confirmed these observations by showing continuous, multi-lobed nuclei (*Figure 2E,F*). BM macrophages also displayed numerous pseudopods emanating from their surfaces. Macrophage pseudopods (*Figure 2—figure supplement 1B*) could be distinguished from uropods emanating from T-cells (*Figure 2—figure supplement 1A*) in that they were smaller and contained organized cytoskeletal filaments, but lacked membranous organelles (*Figure 2—figure supplement 1A–F*). Uropods were also found on macrophages (*Figure 2—figure supplement 2*), but more rarely than found on BM T-cells. Taken together, EM and IF surveys of BM from HIV-1–infected hu-mice revealed the presence, distribution, and morphological details of a subset of cell populations that could be involved in HIV-1 dissemination.

Using ET to gain ultrastructural insight into mechanisms of virus dissemination within HIV-1–infected BM, we identified three potential dissemination mechanisms.

## Dissemination mechanism #1: Synchronous virion release from the cell surface

Virions budding from the plasma membrane were detected by ET as early as 5 days PI on cells morphologically identified as T-cells (*Figure 3A,B*) and macrophages (*Figure 3C,D*), consistent with larger-volume IF imaging of cleared tissue (*Figure 1*). Interestingly, virions were often seen budding from the surface of T-cells at points corresponding to narrow regions of underlying cytoplasm (*Figure 3B*), suggesting either that virus components and associated host machinery can assemble in a space-limited environment and/or that the morphology of HIV-1–infected cells is dynamic. The finding of budding virions along the plasma membrane of virus-producing cells that were not in direct contact with other cells (common in BM due to its low cell density) suggested that the release of free virions from the plasma membrane is an in vivo HIV-1 dissemination mechanism.

HIV-1 virions budding from the surface of virus-producing cells in the BM often showed a series of 2–4 thin, but electron-dense, bands that circumscribed the constricting bud neck (*Figure 3E*; *Figure 3—figure supplement 1*). We previously used immuno-EM to demonstrate that the CHMP1B and CHMP2A ESCRT-III proteins are present at locations in budding virions where these bands of electron density form (*Ladinsky et al., 2014*). In addition, the location of the bands within the bud neck is consistent with their identification as components of ESCRT-III, which transiently polymerize into spirals at the necks of budding virions to facilitate plasma membrane abscission and virus release (*Sundquist and Krausslich, 2012*) (*Figure 3E*; *Figure 3—figure supplement 1*). Finally, their resemblance to EM structures of polymerized ESCRT-III proteins and similarity to a 'dome' model of ESCRT-mediated membrane scission (*Schöneberg et al., 2017*; *Sundquist and Krausslich, 2012*) (*Figure 3—figure supplement 1A*) further suggests that the bands represent polymerized ESCRT-III spirals (*Johnson et al., 2018*; *McCullough et al., 2015*; *McCullough et al., 2018*; *Sundquist and Krausslich, 2012*).

In surveys of HIV-1 budding profiles emanating from virus-producing BM cells we confirmed that putative ESCRT-III densities were present only during a stage in the budding process when the bud

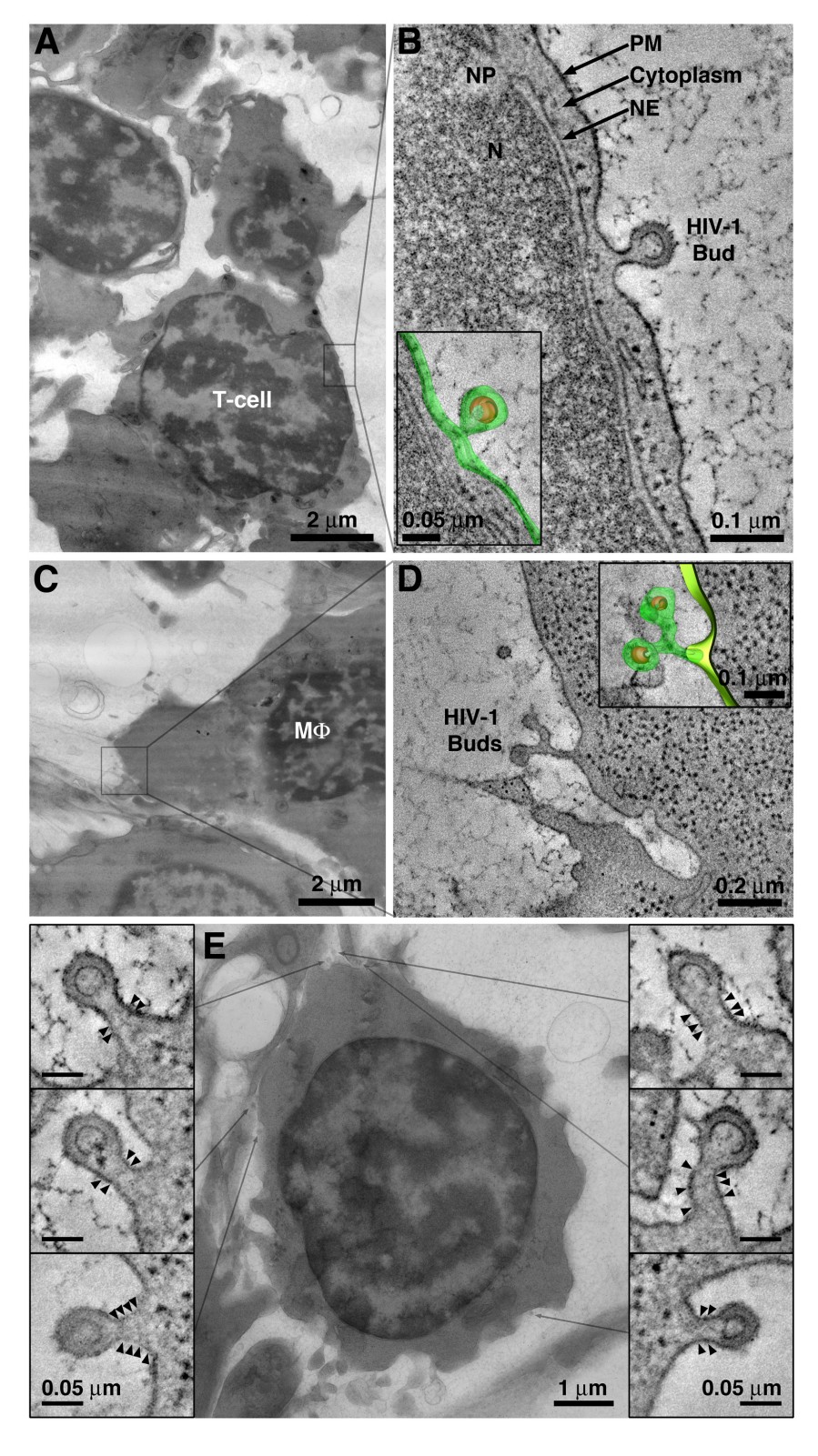

**Figure 3.** Detection of viruses budding from cells in BM. (A) EM overview of a BM region containing a virus-producing T-cell. (B) Tomogram of an HIV-1 budding profile emanating from the surface of a T-cell showing little cytoplasm between the nuclear envelope (NE) and the budding plasma membrane (PM). N = nucleus; NP = nuclear pore. Inset: 3-D model of the HIV-1 bud (green; plasma membrane; orange, immature HIV-1 core). (C) EM overview of a BM region containing a virus-producing macrophage (MΦ). (D) Tomogram showing two HIV-1 budding profiles emanating from a single stalk.
*Figure 3 continued on next page*

*Figure 3 continued*

Macrophages often exhibited multiple (up to 5) viruses emanating from a single stalk. (**E**) Projection EM image showing HIV-1 budding profiles emanating from a virus-producing T-cell in BM (center). Six budding profiles were present on the cell within the volume of the reconstructed tomogram. Each bud had a similar neck diameter and discernable bands of electron density (arrowheads) potentially representing polymerized ESCRT-III fission machinery.

DOI: https://doi.org/10.7554/eLife.46916.007

The following figure supplement is available for figure 3:

**Figure supplement 1.** HIV-1 budding profiles displaying ESCRT-III spirals.

DOI: https://doi.org/10.7554/eLife.46916.008

neck was ~50–80% of the diameter of the budding virion itself (*Figure 3E*; *Figure 3—figure supplement 1B*). The bands were not present when the bud neck was larger or smaller than that ratio, corresponding to earlier or later budding stages, respectively (*Figure 3—figure supplement 1B*). Although ET of tissue samples produces a series of static images, we can use surveys of budding profiles to deduce aspects of a dynamic process such as budding. For example, the observation that the putative ESCRT-III densities are only present on buds that have relatively thick necks suggests that the presence of ESCRT-III spirals connotes a transient, and specific, temporal stage in the HIV-1 budding process. This suggestion is consistent with recent TIRF microscopy studies of individual budding viruses showing ESCRT-III accumulation at a specific stage of HIV-1 Gag accumulation and membrane closure prior to membrane constriction (*Johnson et al., 2018*).

Virus-producing T-cells in BM often showed numerous nascent virions budding from multiple sites along the plasma membrane (*Figure 3E*). Because our ET images represent snapshots of specific time points in the budding process captured by rapid fixation, we infer that the nascent virions represent simultaneous budding events. In such cases, ET showed that all buds on a given cell had similar neck diameters, and that all either showed or did not show presumptive ESCRT-III bands. For example, a virus-producing BM T-cell was found to have six budding profiles on its surface within a 400 nm thick section (*Figure 3E*). ET of each bud showed a similar neck diameter and presumptive ESCRT-III bands circumscribing each bud neck. Nascent virions of different neck diameters or that did not include presumptive ESCRT-III bands (*Figure 3E*, outer panels) were not observed on this cell. These observations suggest that all virions budding from this cell were at a similar stage of egress at the time the BM sample was fixed, supporting the hypothesis that release of virus from virus-producing T-cells in BM occurs in a coordinated or synchronous fashion (*Ladinsky et al., 2014*); that is we infer that if all of the buds present on a region of the cell are showing ESCRT bands, they are all at roughly the same point in the budding process. Furthermore, the absence of buds with ESCRT bands within that same region of the cell implies that the budding events were initiated and proceeded at multiple points within roughly the same definable time window.

## Dissemination mechanism #2: Cell-to-cell transmission of virions attached to T-cell uropods

HIV-1 virions were found associated with, and budding from, the plasma membranes of large uropods that projected from T-cells and contacted nearby cells (*Figure 4A–C*). One or two uropods were typically seen per cell. Uropod plasma membranes usually displayed numerous caveolae-like invaginations (*Figure 4D*), implying that uropods are an active zone for endocytosis and signaling (*Cheng and Nichols, 2016*; *Sánchez-Madrid and Serrador, 2009*). The uropod mode of cell-to-cell contact contrasted with the common contact mode of macrophages, in which numerous pseudopods emanating from a macrophage surface engaged with multiple neighboring cells, sometimes enveloping entire cells (*Figure 2—figure supplement 1B*). ET and 3D modeling of a zone-of-contact between a T-cell uropod and a potential target cell showed a region between two uropod processes that included electron-dense material into which a virion was budding (*Figure 4F,G*). Mature virions were also found within plasma membrane invaginations of virus-producing cells that were contacting potential target cells (*Figure 2—figure supplement 1G*). Cumulatively, these observations describe a potential mechanism for virus dissemination in the BM by which uropods actively facilitate the direct transfer of virions between virus-producing cells and target cells across a virological synapse, which we define as a region of close proximity between a donor cell and a potential target cell, often

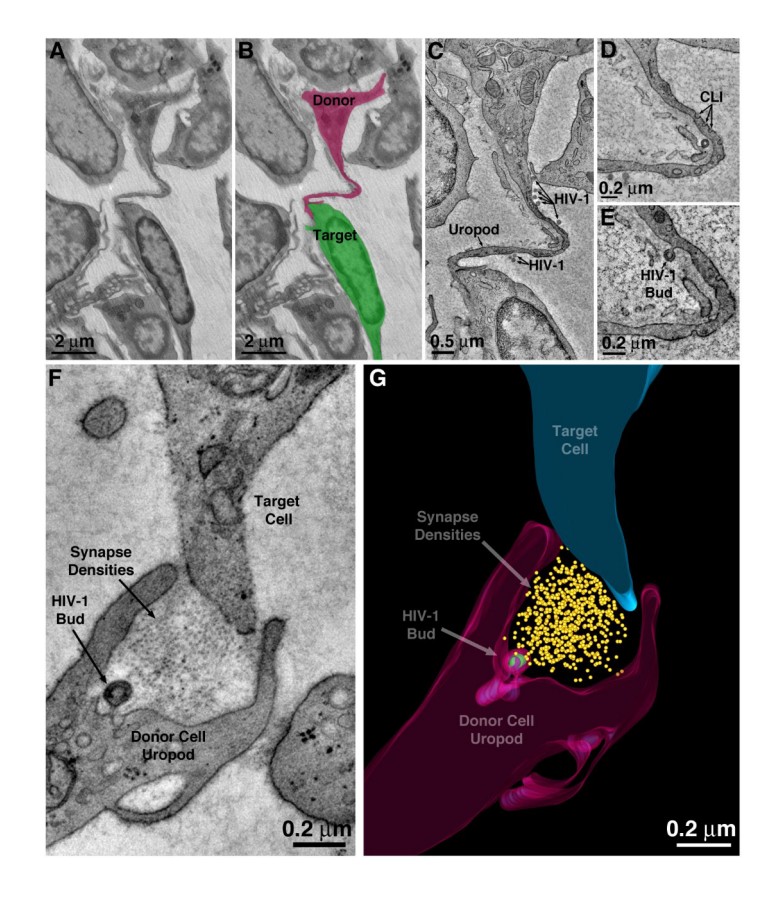

**Figure 4.** HIV-1 transfer via uropod. (**A–B**) Projection EM image of cell-cell contact via uropod in BM (panel A) with modeled donor cell uropod (magenta) and target cell (green) (panel B). (**C**) Tomographic slice of the zone of contact between the donor uropod and target cell. Several free HIV-1 virions are visible along the length of the uropod. (**D**) Caveolae-like invaginations (CLI) along the uropod. (**E**) A virus budding from the plasma membrane of the uropod. (**F**) A virus budding from a uropod in contact with a potential target cell and possibly forming a virological synapse.

DOI: https://doi.org/10.7554/eLife.46916.009

including zones of direct contact, that facilitates transfer of material from the donor cell to the target cell.

## Dissemination mechanism #3: Macrophage phagocytosis of virus-producing T-cells and release of intracellular virions

BM macrophages (identified by hCD68 labeling) associated with HIV-1 p24 were detected by IF microscopy (*Figure 1*), prompting higher resolution analysis by ET to characterize and ascertain their role in virus dissemination. BM macrophages are characterized by having large populations of endocytic compartments, phagosomes and numerous morphologically-complex invaginations of the cell surface (*Sewald et al., 2015*). Although both BM macrophages and neutrophils in BM from BLT humice were indeed polymorphonuclear, neutrophils can be easily distinguished from macrophages by their smaller size and populations of large, evenly dense granules. Furthermore, neutrophils lack extensive, pleomorphic surface invaginations that are common to macrophages and are often enhanced after infection by HIV-1 (*Deneka et al., 2007*; *Graziano et al., 2016*). Such invaginations appeared electron-lucent in BM macrophages (*Figure 2*; *Figure 7—figure supplement 1*).

Cumulatively, ET showed several modes of BM macrophages associated with HIV-1 virions, including phagosomes containing ingested virus-producing cells (*Figure 5*), virions budding from the

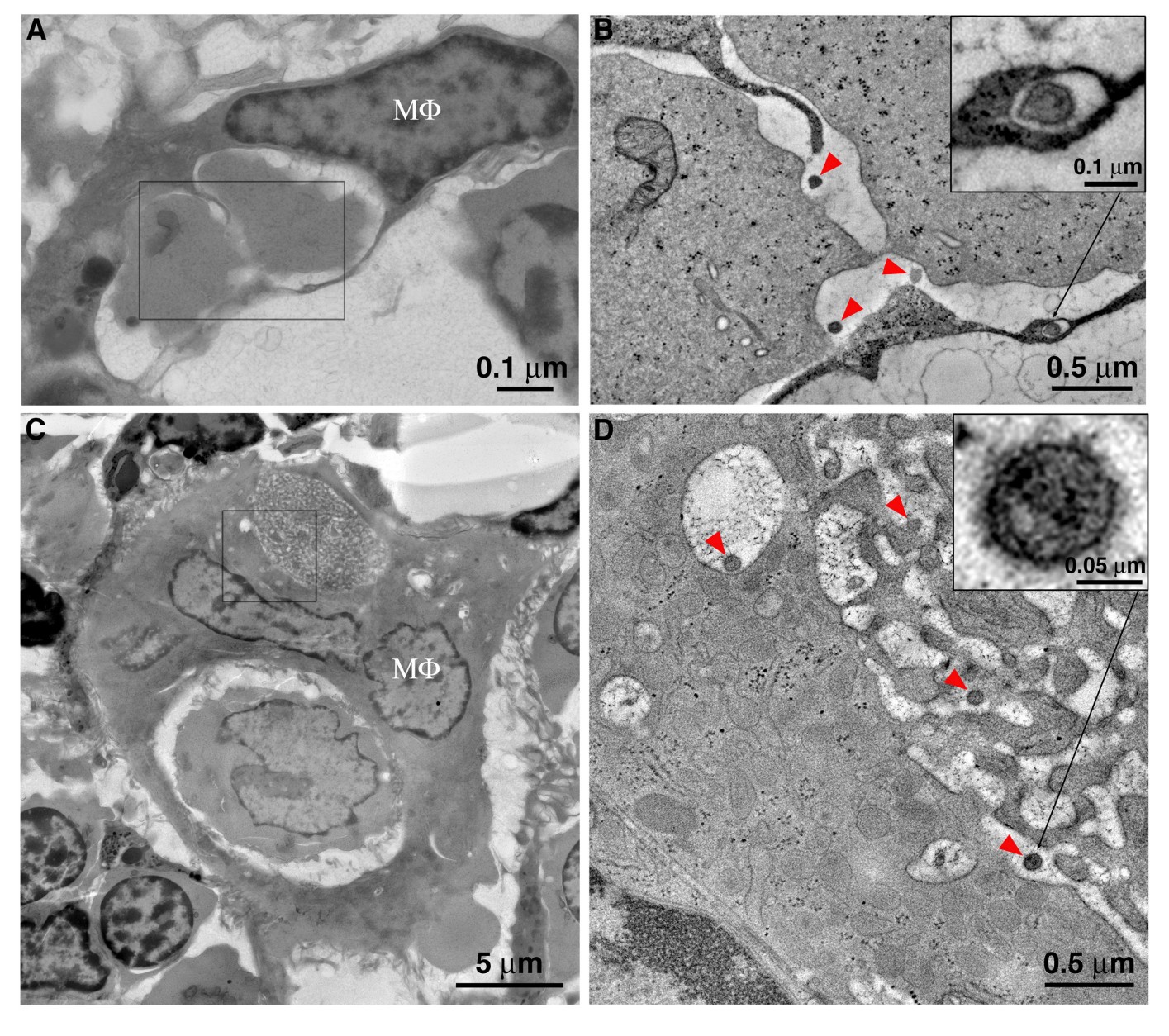

**Figure 5.** Macrophages phagocytose virus-producing cells within BM, and mature HIV-1 survives phagocytic degradation. (**A**) A typical BM macrophage with a partially degraded cell within a phagosome. (**B**) Tomogram detail of the region indicated by the rectangle in A, showing mature HIV-1 virions (red arrowheads) within the phagosome, adjacent to the degrading cells. Inset: Mature HIV-1 virion within an endocytic compartment adjacent to the phagosome. (**C**) An enlarged polymorphonuclear macrophage with two phagocytosed cells at different stages of degradation. The cell in the upper phagosome is nearly completely degraded. (**D**) Tomogram detail of the region indicated by the square in C, showing a portion of the upper phagosome region. Mature HIV-1 virions (red arrowheads) are present within the phagosome and within an adjacent endocytic compartment that is continuous with the phagosome. Inset (upper right): higher magnification detail of an HIV-1 virion, confirming its identity by the presence of a cone-shaped core. Inset (lower left): Detail of a free, immature HIV-1 virion demonstrating an incomplete 'C-shaped' core.
DOI: https://doi.org/10.7554/eLife.46916.010

cell surface (*Figure 3C,D*), virions contained within small intracellular compartments in the cytoplasm (*Figure 6*), and virions contained within surface invaginations (*Figure 7*; *Figure 7—figure supplement 1*). We estimate BM macrophages (as identified morphologically) that did not contain HIV-1 virions outnumbered macrophages associated with virions by ~20:1.

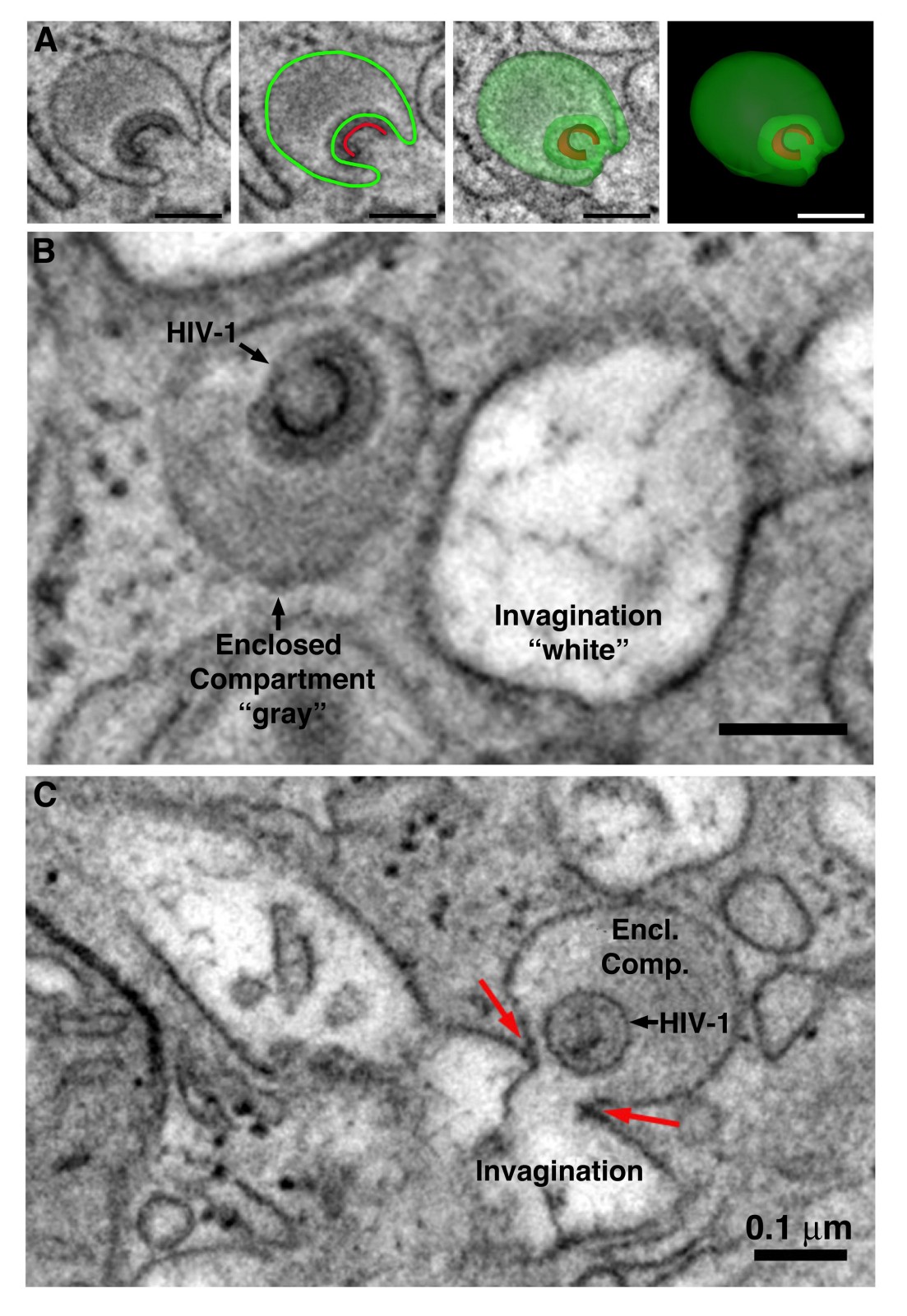

**Figure 6.** Comparison of HIV-1–containing enclosed compartments and virus-containing compartments accessible to the extracellular space. (A) Segmentation of an HIV-1-containing enclosed compartment from a tomographic reconstruction of a BM macrophage. Left to right: Tomographic slice, segmentation contours on a tomographic slice (green: enclosed compartment boundary; red: nascent HIV-1 core) showing that the compartment is completely contained within the section volume and there are no associated structures connecting with the cell surface (*Figure 6—video 1*). (B) HIV-1–
*Figure 6 continued on next page*

*Figure 6 continued*

containing enclosed compartments are distinguished from surface invaginations by their lumenal density (gray for enclosed compartments versus white for surface-accessible compartments) (*Figure 6—video 2*). (C) Tomographic slice demonstrating fusion of an HIV-1–containing enclosed compartment with a surface invagination. Electron density is slightly higher at the point of fusion (red arrows), suggesting diffusion of the compartment's contents and a possible route of HIV-1 dissemination from the macrophage (*Figure 6—video 3*). All scale bars are 0.1 µm.

DOI: https://doi.org/10.7554/eLife.46916.011

The following videos are available for figure 6:

**Figure 6—video 1.** HIV-1 buds into completely enclosed compartments in macrophages.
DOI: https://doi.org/10.7554/eLife.46916.012

**Figure 6—video 2.** Tomography of intracellular compartments from BM macrophages to correlate reduced luminal electron density with access to the extracellular space.
DOI: https://doi.org/10.7554/eLife.46916.013

**Figure 6—video 3.** Fusion of completely enclosed intracellular compartments with surface invaginations as a mechanism for virus dissemination by macrophages.
DOI: https://doi.org/10.7554/eLife.46916.014

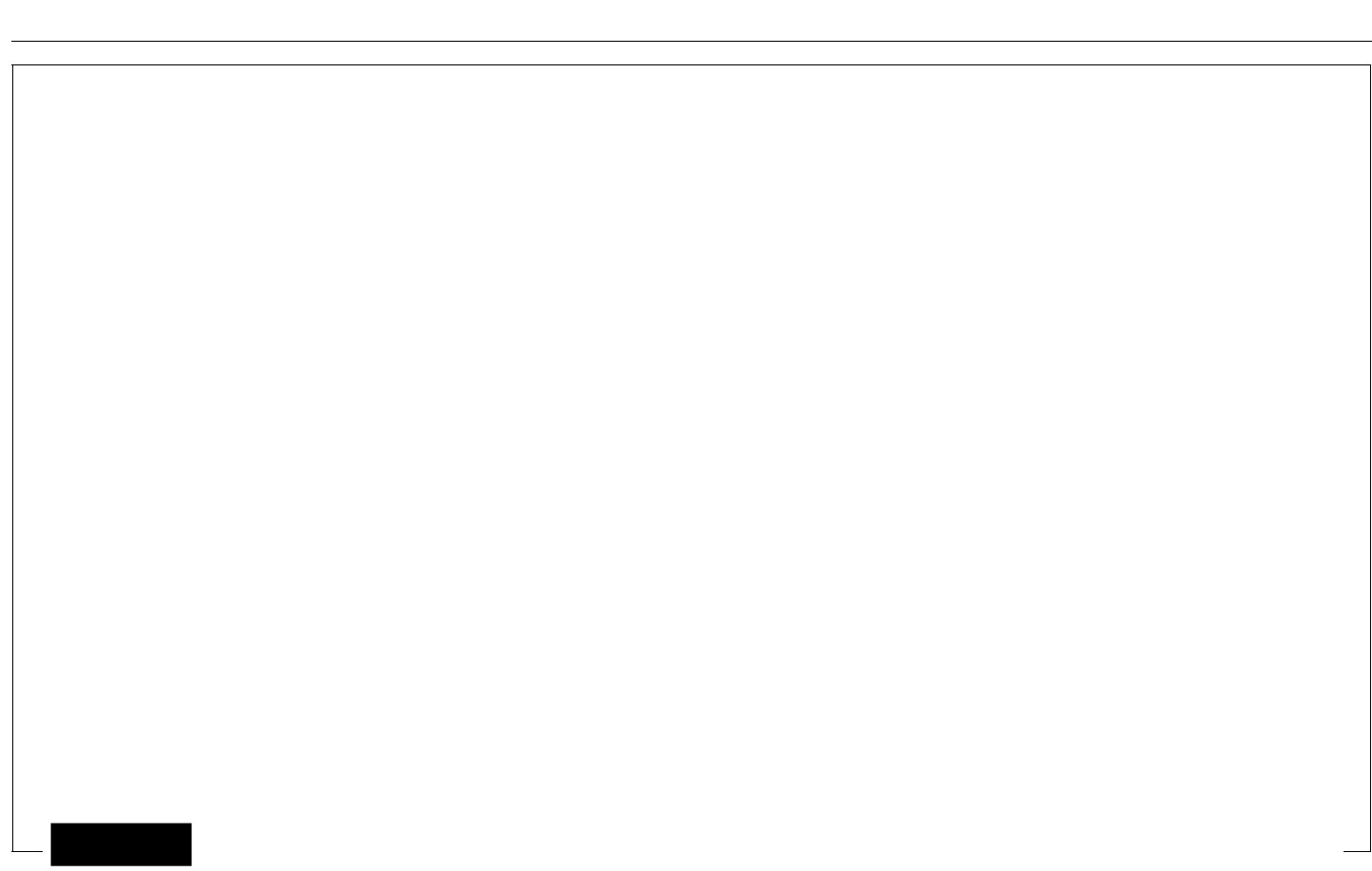

**Figure 7.** Quantification of HIV-1 virions associated with enclosed compartments in a BM macrophage. All HIV-1 particles were located and tabulated within a tomographic reconstruction representing a 5.5 µm x 5.5 µm x 1.5 µm volume of an enlarged, polymorphonuclear BM macrophage. (A) Representative slice from the montaged tomogram, indicating the relative positions of virions budding into enclosed compartments (magenta), free immature (yellow) and mature (green) virions within enclosed compartments, and mature virions within surface invaginations (blue). (B) Gallery of examples of each class of virion from the tomogram.

DOI: https://doi.org/10.7554/eLife.46916.015

The following figure supplement is available for figure 7:

**Figure supplement 1.** HIV-1 buds into, and matures within, enclosed compartments in BM macrophages.
DOI: https://doi.org/10.7554/eLife.46916.016

Montaged projection EM overviews revealed BM macrophages in the process of phagocytosing T-cells (*Figure 5*) and ET often showed mature and immature free HIV-1 virions within macrophage phagosomes alongside engulfed cells at varying degrees of degradation. Classification of free virions as mature or immature was based upon the presence of a cone-shaped core (mature) (*Figure 5B,D*; upper right inset) or incomplete 'C-shaped' core (immature) (*Figure 5D*; lower left inset). Virions were also found in early endosomal compartments adjacent to and/or continuous with the phagosome (*Figure 5C,D*), suggesting a pathway for transfer of HIV-1 from ingested T-cells into a macrophage.

Previous studies of cultured macrophages and cells within GALT described intracellular virus-containing compartments, but these were continuous with thin channels that allowed direct access to the extracellular space (*Bennett et al., 2009*; *Deneka et al., 2007*; *Ladinsky et al., 2014*). In contrast, this current study detected a class of compartments that were fully enclosed and showed no steady-state continuity with extracellular space. Tomography and 3D modeling of these compartments and the adjacent cellular volume demonstrated a clearly-distinguishable continuous membrane surrounding the compartment in all dimensions, with no association with other structures that linked them to extracellular space (*Figure 6A,B*; *Figure 6—video 1*). HIV-1 virions were observed assembling at, budding into, and maturing within these compartments (*Figure 7*).

Enclosed compartments containing virions were distinguished from pleomorphic surface invaginations by the electron densities of their respective lumens because the content of the enclosed compartments was denser and thereby appeared darker by EM (*Figure 6A,B*). By contrast, the lumens of invaginations were mostly devoid of electron dense content and appeared comparatively lighter (*Figure 6B,C*). The increased density of enclosed compartments likely resulted from retained biomolecules that were otherwise lost by fluid exchange from compartments with access to the extracellular space during tissue fixation. Tomography of BM macrophages confirmed that compartments with lighter lumens were continuous with the cell periphery and therefore open to extracellular space (*Figure 6—video 2*).

Although virions were observed budding from the plasma membranes of BM macrophages (*Figure 3C,D*), ET more frequently revealed examples of virions budding into completely-enclosed compartments with dark lumens (*Figure 7B*; *Figure 7—figure supplement 1*). These results contrast with previous studies in cultured cells demonstrating that HIV-1 buds into internal regions of membrane contiguous with the plasma membrane rather than into endocytic compartments (*Gaudin et al., 2013*). The discovery of virions budding into completely enclosed compartments of BM macrophages raises the question of whether budding into an enclosed compartment represents a dead end for a nascent virion. In an example demonstrating that budding into an enclosed intracellular compartment could result in eventual virus release from a macrophage, an enclosed compartment with a dark lumen containing a mature virus particle was observed in the process of fusing with a surface-accessible compartment with a lighter lumen (*Figure 6C*; *Figure 6—video 3*), suggesting a pathway by which intracellular viruses can access and subsequently disseminate into extracellular space.

In order to classify HIV-1 virions associated with enclosed compartments, a large volume of a polymorphonuclear BM macrophage (5.5 µm x 5.5 µm x 1.5 µm) was imaged by ET (*Figure 7*). Within this volume, we classified virions as either budding into enclosed compartments, free immature virions in enclosed compartments, mature virions in enclosed compartments, or mature virions in surface invaginations (*Figure 7A,B*). This volume revealed 357 virions (budding, immature, and mature) located within intracellular locations of a virus producing macrophage. The large number (98) of observed budding virions suggested prolific budding into enclosed compartments. The number of free mature virions within enclosed compartments outnumbered free immature virions (216 versus 17), suggesting rapid maturation. Relatively few mature virions (8) were found within surface-accessible invaginations, suggesting that the transfer of virions by fusion of enclosed compartments with surface invaginations is infrequent and slow.

## Discussion

Light and electron microscopy of HIV-1–infected cultured cells have revealed important insights into the HIV-1 life cycle and mechanisms of virus transmission (*Bennett et al., 2009*; *Bracq et al., 2017*; *Deneka et al., 2007*; *Folks et al., 1988*; *Martin et al., 2010*; *Sougrat et al., 2007*), but cultured

cells cannot replicate complex interactions encountered in vivo by viruses in the context of functioning immune cell populations within tissues. Imaging of HIV-1–infected lymphoid tissues is establishing the importance of tissues in virus dissemination, with each tissue providing insights into HIV-1 pathogenesis (*Carias et al., 2013*; *Kieffer et al., 2017a*; *Kieffer et al., 2017b*; *Ladinsky et al., 2014*; *Law et al., 2016*; *Li et al., 2005*; *Murooka et al., 2012*; *Reinhart et al., 1997*; *Sewald et al., 2015*). HIV-1 infection of BM cells can cause hematopoietic defects (*Alexaki and Wigdahl, 2008*; *Allen and Dexter, 1984*; *Folks et al., 1988*; *Mandell et al., 1995*; *Redd et al., 2007*; *Thiebot et al., 2001*; *Yamakami et al., 2004*), yet relatively little is known about mechanisms of HIV-1 dissemination in BM and their influence on downstream aspects of disease. Here we utilized a multiscale 3D tissue imaging approach to probe mechanisms of virus dissemination in BM from HIV-1-infected BLT hu-mice using IF of large volumes of clarified tissue and ET of smaller tissue volumes at higher resolution.

Although directly-comparative imaging of infected human or NHP BM would be an informative addition to our study, we note that the high-resolution ET imaging studies described here required extraction of BM by direct removal from a long bone (see Materials and methods), which is not a possible sample preparation method from living humans or NHPs. BM can be extracted from living subjects using needle aspiration, but in direct comparisons of macaque BM extracted by needle aspiration versus removal from a long bone, we could not locate physiologically-relevant regions that were distinct from mature red blood cells in the needle aspiration samples because they contained contaminating blood from the vasculature introduced during the procedure (M.S.L and P.J.B., unpublished observations). Nonetheless, the imaging studies in BLT hu-mouse BM provided structural insights into mechanisms of HIV-1 dissemination in BM and potentially other tissues including: (*i*) virus release from the plasma membrane into extracellular space, (*ii*) cell-associated transmission of virions attached to T-cell uropods, and (*iii*) transfer of virus into macrophages from ingested virus-producing T-cells.

Many cell types permissive to HIV-1 reside in BM, including CD4+ T-cells, macrophages, and dendritic cells (*Alexaki and Wigdahl, 2008*; *Allen and Dexter, 1984*; *Folks et al., 1988*). CD4+ T-cells make up the largest lymphocyte subset in human BM (*Oetjen et al., 2018*), and reduction of CD4+ T-cell levels in BM is associated with many infections and cancers (*Di Rosa and Pabst, 2005*). Monocyte/macrophages also make up a substantial cellular component of human BM and are integral for tissue homeostasis, HSC maintenance, and immune surveillance (*Kaur et al., 2017*). We detected virus-producing T-cells and macrophages in HIV-1–infected BLT hu-mouse BM as early as 5–10 days PI by IF and EM, a timeframe similar to that detected in NHP BM (*Mandell et al., 1995*). Additionally, light microscopy of BM from HIV-1–infected hu-mice revealed hCD4+ T-cell depletion over time. These results are consistent with NHP studies (*Hoang et al., 2019*) and human patients with AIDS (*Harris et al., 1990*), which together support the use of hu-mice as a model to probe mechanisms of HIV-1 dissemination in BM. Vascularization supporting migration of cells to and from BM (e. g., see *Figure 2A*), combined with trafficking of HIV-1–permissive cells, could be responsible for the rapid seeding of this tissue with virus-producing cells detected by ET. Because many immune cell types in BM are permissive to HIV-1 and can potentially migrate to other lymphoid tissues to take up permanent tissue residence (*Alexaki and Wigdahl, 2008*; *Gorantla et al., 2006*), we speculate that migration of HIV-1–infected cells to and from BM is a potential mechanism for virus dissemination to lymphoid tissues throughout the body. Studies in NHPs showed increased levels of monocyte/macrophages released from bone marrow upon infection with SIV/SHIV (*Hasegawa et al., 2009*). This scenario would allow seeding of distant lymphoid tissue sites with virus-producing cells to enhance virus production and systemic spread early during HIV-1 infection.

Although virus-producing cells were detected in BM within 5 days after infection, fewer such cells were found in BM throughout the duration of the infection as compared to virus-producing cells from infected hu-mouse GALT at similar time points (*Kieffer et al., 2017b*; *Ladinsky et al., 2014*). This could be due to the lower reported density of T-cells in BM (1–3%) (*Bonomo et al., 2016*; *Monteiro et al., 2005*) as compared with that of GALT of hu-mice (~6%) (*Kieffer et al., 2017b*) or of human patients (~13%) (*McElrath et al., 2013*). IF and EM analysis of BM revealed hCD68+ macrophages as a major cell population producing HIV-1 p24 (*Figures 1* and *3*) which parallels NHP studies showing monocytes/macrophages to be an early and important target of SIV in BM (*Kitagawa et al., 1991*; *Mandell et al., 1995*). IF revealed a sizeable population of p24+ cells that did not stain as hCD68+ or hCD4+ (*Figure 1E*), consistent with CD4 downregulation by HIV-1–

infected T-cells, or cell-free viruses. An additional possibility is this cell population represents CD4-/CD68- BM myeloid suppressor cells, which were shown to be infected by SIV in vivo (*Sui et al., 2017*).

HIV-1 can be transmitted through infection by cell-free virus or through cell-to-cell contact, with the latter mechanism being more efficient in cultured cells (*Carr et al., 1999*; *Chen et al., 2007*; *Dimitrov et al., 1993*; *Jolly et al., 2004*; *Phillips, 1994*). In tissues, the prevalence and efficiency of these two mechanisms of virus transmission is less well understood, partly because of the complex and dynamic environment that HIV-1 encounters the presence of an immune system and multiple cell types organized into tissues (Reviewed in *Uchil et al., 2019*). Intravital imaging studies revealed cell-to-cell transmission of HIV-1 in tissues of hu-mice (*Murooka et al., 2012*; *Sewald et al., 2012*); however, the resolution of LM makes it difficult to definitively assign virion fluorescence as representing cell-free versus cell-associated virions. In this study, we used ET to directly visualize cell-associated and cell-free virions within BM during active HIV-1 infection in hu-mice. One important observation from ET of BM was that examples of both cell-free and cell-associated virions were readily detected, suggesting that HIV-1 transmission in vivo can occur by both mechanisms.

In previous ET studies of more densely-packed cells in hu-mouse tissues such as GALT and spleen, we found examples of potential cell-to-cell spread of virions as well as abundant free virions (*Kieffer et al., 2017b*; *Ladinsky et al., 2014*). Because the overall cell density in BM is reduced compared to other lymphoid tissues (*Figure 2A*; *Figure 2—figure supplement 1*), we speculate that virus-producing cells contacting other cells are more rare in BM than in densely-packed tissues, but potentially more directed due to the lower density of cells. Furthermore, in contrast to our previous ET studies of HIV-1-infected BLT hu-mouse GALT, which revealed hundreds of mature virions localizing between cells (*Ladinsky et al., 2014*), we detected fewer cell-free virions in hu-mouse BM. One interpretation for this difference is that the lower cell density of BM compared with other tissues allows cell-free virus to more rapidly disseminate from virus-producing cells as opposed to being trapped in pools between densely-intercalated cells in GALT or spleen (*Kieffer et al., 2017b*; *Ladinsky et al., 2014*). These contrasting results highlight the importance of tissue-specific characteristics (e.g., cell densities, cell populations, architecture, function) influencing HIV-1 transmission in different lymphoid tissues and that HIV-1 transmission does not occur by a singular mechanism in all tissues.

ET analysis allowed identification of different stages of virus egress by characterizing thin bands of electron density circumscribing virus bud necks. These presumably represent components of the host ESCRT-III machinery that are involved virus budding (*Sundquist and Krausslich, 2012*) (*Figure 3E*; *Figure 3—figure supplement 1*), as previously demonstrated by immuno-EM (*Ladinsky et al., 2014*). We detected examples of budding virions from individual cells that were all at similar stages of egress as indicated by the putative ESCRT bands and by bud necks widths that were ~50–80% the diameter of the nascent virion itself. Budding virions with larger or smaller ratios of neck-to-virus diameter, corresponding to earlier or later stages of budding, respectively, did not show these bands, suggesting transient formation of this structure at a specific stage of virus release. These observations are consistent with TIRF microscopy studies of individual budding virions that showed ESCRT-III accumulation at a specific stage of HIV-1 Gag accumulation and membrane curvature prior to membrane constriction (*Johnson et al., 2018*). Taken together, these results support a model of synchronized release of virions from a cell in which spirals of ESCRT-III transiently polymerize at the necks of budding virions at a specific stage of Gag accumulation and membrane curvature to facilitate plasma membrane abscission and virus release.

A striking observation revealed by ET was the prevalence of BM macrophages engulfing virus-producing cells, with examples at different stages of engulfment and degradation, demonstrating containment of intact, virus-producing cells in addition to individual virions within intracellular macrophage compartments (*Figures 5–7*; *Figure 6—videos 1–3*. These results suggested that macrophages in BM actively engulfed and degraded T-cells that were producing virus and could represent an example in lymphoid tissue similar to results in cultured macrophages describing removal of virus-producing T-cells via efferocytosis (phagocytic clearance of apoptotic cells) (*Baxter et al., 2014*; *Chua et al., 2018*) and to reports of macaque macrophages containing phagocytosed SIV-infected CD4+ T-cells (*DiNapoli et al., 2017*). Our results also address in vitro fluorescence microscopy and EM studies of HIV-1–infected human CD4+ T-cells cultured with uninfected human macrophages that showed fusion of infected T-cells and macrophages to form multinucleated giant cells

(*Bracq et al., 2017*); similar phenomena were also reported through EM analysis of human patient samples (*Orenstein and Wahl, 1999*). By contrast, we did not observe examples of macrophage–T-cell fusion, although we found many examples of macrophages ingesting cells resembling T-cells in BM. In addition, we showed by 3D imaging that macrophages that appeared to include multiple nuclei in individual IF optical sections or EM projection images actually contained single, polymorphous nuclei with connections between individual nuclear lobes (*Figure 2D*). However, EM of macrophages in the process of engulfing cell(s) included the additional nuclei from the ingested cell itself, and ET showed these nuclei to be physically separated from the nucleus and cytoplasm of the macrophage by a membrane-enclosed space (*Figure 5*). For example, *Figure 5C* shows a macrophage that completely engulfed two other cells, each delineated by intact membranes or phagosomes. By light microscopy this could be interpreted as a single cell containing multiple distinct nuclei within the cytoplasm.

We detected BM macrophages producing and harboring large numbers of virions within intracellular locations. In one example, 357 virions were detected within a 5.5 µm x 5.5 µm x 1.5 µm volume of a BM macrophage. Macrophages are larger than T-cells and extrapolating these results to the approximate volume of the macrophage imaged (~2800 um$^3$) and subtracting half of the volume as nucleus not containing virions (~1400 um$^3$), an individual BM macrophage could harbor >11,000 virions. These virions could represent a potential virus reservoir as macrophages are long-lived, motile, and frequently interact with target cells. Virus-containing intracellular compartments have been observed in cultured macrophages (*Bennett et al., 2009*; *Deneka et al., 2007*) and in cells from GALT and spleen (*Kieffer et al., 2017b*; *Ladinsky et al., 2014*). However, in previous cases, the apparently intracellular compartments were directly continuous with the extracellular space by means of thin channels. Here, we found examples of virus-containing intracellular compartments that were enclosed with no continuity with the extracellular space (*Figure 6A*, *Figure 7—figure supplement 1*; *Figure 6—video 1*). HIV-1 was observed budding into these compartments and at different stages of maturation within them, suggesting that they serve as sites for virus assembly and associated host-cell derived budding machinery. The completely-enclosed compartments in BM macrophages typically contained only 1–3 virions in the form of nascent buds, immature free virions, or mature free virions. This result contrasts with virus-containing compartments in cultured macrophages and in cells within GALT and spleen tissue, which contained ≥20 (GALT and spleen) and ≥50 (cultured cells) virions. Budding into a completely-enclosed intracellular compartment seems unproductive for a virus; however, we observed examples of fusion between a completely-enclosed compartment and a compartment with access to the extracellular space (*Figure 6*; *Figure 6—video 3*), supporting a route by which viruses could be produced, stored, and released by macrophages in tissues.

Cumulatively, our studies reveal macrophages to be actively involved HIV-1 dissemination in BM, suggesting the intriguing possibility that virus-producing BM macrophages could support systemic virus spread to distant tissue locations as they differentiate into tissue-resident macrophages and migrate to distant lymphoid sites. The ability of BM macrophages to produce and harbor virus early after infection could function as a virus reservoir with the potential to promote virus dissemination to distant tissue locations.

## Materials and methods

**Key resources table**

| Reagent type (species) or resource | Designation | Source or reference | Identifiers | Additional information |
|---|---|---|---|---|
| NOD.Cg-*Prkdc*$^{scid}$ *Il2rg*$^{tm1Wjl}$/SzJ (*Mus musculus*) | NSG | UCLA CFAR Humanized Mouse Core laboratory | | |
| NFNSX (*HIV-1*) | NFNSX | *O'Brien et al., 1990* | | |

*Continued on next page*

*Continued*

| Reagent type (species) or resource | Designation | Source or reference | Identifiers | Additional information |
|---|---|---|---|---|
| Biological sample (*Homo sapiens*) | Human fetal thymus and liver | Advanced Bioscience Resources (ABR) | | Freshly isolated from de-identified human fetus |
| Antibody | anti-human CD34+ (Mouse monoclonal) | Miltenyi Biotec | Catalog # 130-100-453 | FACS (100 μL for up to 1e8 cells. |
| Antibody | anti-human CD45-eFluor 450 (mouse monoclonal, clone HI30) | eBiosciences | Catalog # 48-0459-42 | 5 μL (0.5 μg)/test |
| Antibody | anti-human CD3-APC H7 (mouse monoclonal, clone SK7) | BD Pharmingen | Cat# 340440 | 10 μL / 100 μL test |
| Antibody | anti-human CD4-APC (mouse monoclonal, clone OKT4) | eBiosciences | Catalog # 17-0048-42 | 5 μL (0.06 μg)/test |
| Antibody | anti-human CD8-PerCP Cy5.5 (mouse monoclonal, clone SK1) | BioLegend | Catalog # 344710 | 5 μl per million cells in 100 μl |
| Antibody | anti-human CD19-Brilliant Violet 605 (mouse monoclonal, clone HIB19) | BioLegend | Catalog # 302244 | 5 μl per million cells in 100 μl |
| Antibody | anti-mouse FcR (CD16/32); (rat monoclonal, clone 93) | BioLegend | Catalog # 101302 | 1:100 |
| Antibody | anti-human CD68 (mouse monoclonal, clone PG-M1 | Dako | Catalog # M0876 | 1:200 |
| Antibody | anti-p24 (HIV-1); (mouse monoclonal, clone Kal-1) | Dako | Catalog # M0857 | 1:200 |
| Antibody | anti-human CD4 (mouse monoclonal, clone 4B12) | Dako | Catalog # M7310 | 1:200 |
| Antibody | anti-p24 (HIV-1); (goat polyclonal) | Creative Diagnostics | Catalog # DPATB-H81692 | 1:200 |
| Antibody | Anti-goat IgG Alexa Fluor 633 (Donkey polyclonal) | Life Technologies | Catalog # A21082 | 1:1000 |
| Sequence-based reagent | HIV-1 NFNSX Gag Forward | This paper | PCR primer | 5′-CCCTACCAGCATT CTGGACATAAG-3′ |
| Sequenced-based reagent | HIV-1 NFNSX Gag Reverse | This paper | PCR primer | 5′-GCTTGCTCGG CTCTTAGAGTT-3′ |
| Sequenced-based reagent | HIV-1 NFNSX Gag qRT-PCR Probe | This paper | qRT-PCR Probe | 5′-FAM-ACA AGGACCA AAGGAACCC TT-BHQ1-3′ |

*Continued on next page*

*Continued*

| Reagent type (species) or resource | Designation | Source or reference | Identifiers | Additional information |
| --- | --- | --- | --- | --- |
| Software, algorithm | FlowJo | TreeStar software | V10 | |
| Software, algorithm | Fiji (Image J) | *Schindelin et al., 2012* | | |
| Software, algorithm | Imaris (Bitplane) | Oxford Instruments | V8.4–9.2 | |
| Software, algorithm | IMOD | (*Kremer et al., 1996*; *Mastronarde, 2008*) | | |

## Human CD34+ HSPC and fetal tissue

Human fetal thymus and liver were obtained from Advanced Bioscience Resources (ABR). The UCLA institutional review board (IRB) determined these tissues are not human subjects and IRB review is not required since fetal tissues were obtained without patient identifying information from deceased fetuses. Human CD34+ HSPCs were isolated from fetal liver using magnetic beads conjugated with anti-hCD34+ monoclonal antibodies (CD34 MicroBead Kit UltraPure, human, Miltenyi Biotec) and an AutoMACS pro separator (Miltenyi Biotec). A human thymus from the same donor was used for generation of BLT mice.

## Generation and infection of BLT humanized mice

NOD.Cg-*Prkdc^scid^ Il2rg^tm1Wjl^*/SzJ (NSG) mice were used to generate BLT mice. They were maintained at the UCLA CFAR Humanized Mouse Core laboratory in accordance with a protocol approved by the UCLA Animal Research Committee (UCLA ARC # 2007-092-41A). Experiments conformed to all relevant regulatory standards. BLT mice were generated as previously described (*Melkus et al., 2006*). NSG mice (8 week-old) were myeloablated by the total body irradiation at a dose of 125 cGy. 24 hr later, each mouse was implanted with a piece of human fetal thymus and liver under the kidney capsule. In addition, each mouse was injected with $5 \times 10^5$ FL-CD34$^+$ using a 27-gauge needle via retro-orbital vein injection.

## HIV-1 production and infection of animals

CCR5-tropic HIV-1$_{NFNSX}$ (*O'Brien et al., 1990*) stocks were prepared by a calcium phosphate plasmid DNA transfection method. BLT mice were injected with HIV-1$_{NFNSX}$ (200 ng of p24 Gag) via the retro-orbital vein plexus using a 27-gauge needle.

## HIV-1 viral load assay

Levels of HIV-1 RNA in plasma of infected BLT mice were determined by a RT-PCR assay. Plasma was separated from peripheral blood and stored at −80°C until use. Viral RNA was isolated with a QIAamp viral RNA mini kit (QIAGEN). The RNA was eluted in 25 µl of RNase-free water and 5 µl of elution was applied for qRT-PCR using iScript One-step RT-PCR kit (Bio-Rad laboratories) with following primers/probe specific to HIV-1$_{NFNSX}$ Gag region. Primer Sequence 1: 5'-CCCTACCAGCATTCTGGACATAAG-3', Primer Sequence 2: 5'-GCTTGCTCGGCTCTTAGAGTT-3' Probe: 5'-FAM-ACAAGGACCAAAGGAACCCTT-BHQ1-3'. With these primers/probes, HIV-1 RNA can be quantitatively detected within a range of $10^3$ copies to $10^8$ copies/ml.

## Flow cytometry

Peripheral blood and BM samples were harvested via the retro-orbital vein plexus at the time of euthanasia. Plasma was removed from peripheral blood by centrifugation and the cell fraction was used to stain with antibodies for 30 min at 4 °C. After the staining, cells were treated with red blood cell lysis (RBCL; 4.15 g of NH$_4$Cl, 0.5 g of KHCO$_3$, and 0.019 g of EDTA in 500 mL of H$_2$O) buffer for 10 min and washed with FACS buffer (2% fetal calf serum in phosphate-buffered saline [PBS]). BM samples collected from femurs and spine were finely minced into small fragments and resuspended

in 5 mL of FACS buffer. The BM cell samples were filtered through a 70 µm cell strainer, cells were washed in FACS buffer, resuspended in RBCL buffer for 10 min, and washed again with FACS buffer. Processed cells from peripheral blood and BM were stained with monoclonal antibodies to human CD45-eFluor 450 (HI30:eBiosciences), CD3-APC H7 (SK7:BD Pharmingen), CD4-APC (OKT4:eBiosciences), and CD8-PerCP Cy5.5 (SK1:BioLegend), and CD19-Brilliant Violet 605 (HIB19: Biolegend). Stained cells were fixed with 1% formaldehyde in PBS and examined with Fortessa flow cytometers (BD Biosciences). The data were analyzed by FlowJo V10 (TreeStar) software.

## Tissue preservation

Upon necropsy, lymphoid tissues were isolated from sacrificed animals, immediately rinsed in ice cold cacodylate buffer (5% sucrose in 0.1M sodium cacodylate trihydrate) and preserved in fixative for LM (8% paraformaldehyde, 5% sucrose in 0.1M sodium cacodylate trihydrate) or EM (1% paraformaldehyde, 3% Glutaraldehyde, 5% sucrose in 0.1M sodium cacodylate trihydrate)) as previously described (*Kieffer et al., 2017b*; *Ladinsky et al., 2014*).

## Passive bone clearing

Entire fixed mouse femurs and sternums were cleared based on the PACT-deCAL and Bone CLARITY methods (*Greenbaum et al., 2017*; *Treweek et al., 2015*). Briefly, fixed BM samples were demineralized in 10% EDTA in PBS at 4 °C for 2–3 weeks with daily exchanges of fresh buffer. Samples were embedded in a hydrogel containing 4% acrylamide and 0.25% thermoinitiator (VA-044, Wako Chemicals). Samples were delipidated with 8% SDS in 0.01 M PBS (pH 7.4) for 7–14 days with constant rocking at 37 °C until visually transparent and clearing was not progressing. SDS was exchanged daily. Samples were washed in 0.01 M PBS (pH 7.4) for 24 hr. at room temperature with at least five buffer exchanges. Samples were decolorized with 25% aminoalcohol (*N,N,N',N'*-tetrakis (2-hydroxypropyl)ethylenediamine) in 0.01 M PBS (pH 7.4) for ~7 days at 37 °C with daily buffer exchanges until tissue color did not reduce further. Refractive index matching solution (RIMS) containing 95% Histodenz (Sigma) in 0.01 M PBS (pH 7.4) was used to immerse samples for at least 16 hr prior to autofluorescence imaging.

## Immunostaining of cleared BM samples

For sternum samples, a vertical central channel of BM along the length of the sternum was visible and slightly darker than the rest of the sample after tissue decolorization. ~2 mm horizontal sections through the central channel of BM were cut from the length of the sternum in order to enhance antibody penetration into the tissue during immunostaining. Femur samples were cut into two pieces and pierced with a 33-gauge insulin syringe (Millipore-Sigma) in 5–10 locations along the length of the sample to promote antibody penetration. Cleared samples were rinsed 3 times in 0.01 M PBS (pH 7.4) for 30 min each, blocked overnight in 0.01 M PBS (pH 7.4) containing 4% fetal bovine serum, 0.1% Tween-20, 0.01% sodium azide, and a 1:100 dilution of rat anti-mouse FcR (CD16/32; Biolegend). Samples were incubated for 3–5 days in blocking buffer (lacking rat anti-mouse FcR antibody for the remaining protocol) containing primary antibodies diluted 1:200. Samples were washed five times with wash solution (0.1% Tween-20% and 0.01% sodium azide in 0.01 M PBS pH 7.4) over the course of one day and incubated with fluorophore-conjugated secondary antibodies (Invitrogen) diluted 1:1000 in blocking buffer. In certain instances, primary antibodies were conjugated to a fluorophore using antibody labeling kits (Biotium Inc), and the secondary antibody staining step was omitted. After immunostaining, samples were washed five times in wash buffer over one day, stained with DAPI in wash buffer for 10 min, and washed 3 times for 10 min prior to immersion in RIMS overnight for sample mounting and imaging. Negative controls to ensure specific staining included imaging uninfected tissues, unstained tissues, and tissues stained with secondary antibodies only at equivalent laser intensities between controls and test samples.

## Confocal microscopy

Immunostained samples were mounted in RIMS between two No.0 coverslips (Electron Microscopy Sciences) separated with 0.5 mm adhesive/adhesive silicone isolators (Electron Microscopy Sciences). A LD LCI Plan-Apochromat 25 × 0.8 NA Imm Corr DIC M27 multi-immersion objective (w.d. 0.57

mm) with glycerol was used to capture images on Zeiss LSM-710, 800, and 880 confocal microscopes.

## Light Sheet Fluorescence Microscopy

Z-stacks with a 1–3 mm step size were acquired using a La Vision Ultra II light sheet fluorescence microscope equipped with an Olympus MVX-10 Zoom body (0.63–6.3X magnification), Olympus MVPLAPO 2x Dry lens (NA 0.50), 10 mm working distance dipping cap, and Andor Neo sCMOS camera. Cleared samples were attached to a spiked sample holder and incubated overnight in an imaging cuvette filled with RIMS. Prior to immunostaining, Z-stacks of endogenous autofluorescence at 488 nm were acquired in order to generate a 3D reference volume for the entire tissue sample. After immunostaining, samples were remounted as described above and Z-stacks were acquired at 488 nm, 546 nm, and 633 nm.

## Image processing

Individual confocal Z-slices were processed using the Fiji (ImageJ) platform (*Schindelin et al., 2012*). Images were thresholded manually and smoothed to reduce background autofluorescence for visualization. Zoomed regions of Individual cells were captured in order to compare characteristic morphologies and sizes of individual stained cell populations.

LSFM datasets were segmented with the Imaris software suite to create a surface models that contained bone and BM instead of muscle in cleared tissues, and generate a 3D reference volume for the sample prior to immunostaining when higher magnification images were acquired. Z-stacks were manually thresholded to minimize background autofluorescence within the volume. Gamma levels were manually adjusted for the purpose of visualizing individual cells with large differences in fluorescence intensity within the same volume and avoid oversaturation. Surface models were segmented using Imaris and individual cell populations within acquired LSFM volumes were automatedly quantified. Three volumes of greater than 0.5 mm$^3$ were quantified from 1 to 2 samples for each time point.

## TEM sample preparation

BM for TEM studies was extracted from the sternum of HIV-1–infected BLT mice. Sternums were removed from the animals and immediately fixed for at least 1 hr in cold (4°C) 3% glutaraldehyde, 1% paraformaldehyde, 5% sucrose in 0.1M sodium cacodylate trihydrate to render the tissues safe for further processing. Sternums were placed in fresh cacodylate buffer and the marrow dissected from the bone core using #5 forceps and a microsurgical scalpel. Portions of BM were placed into brass planchettes (Type A; Ted Pella, Inc, Redding, CA) prefilled with 10% Ficoll in cacodylate buffer, covered with the flat side of a Type-B brass planchette and rapidly frozen with a HPM-010 high-pressure freezing machine (ABRA Fluid AG, Widnow, Switzerland). The vitrified samples were transferred under liquid nitrogen to cryotubes (Nunc) containing a frozen solution of 2.5% osmium tetroxide, 0.05% uranyl acetate in acetone. Tubes were loaded into an AFS-2 freeze-substitution machine (Leica Microsystems) and processed at −90°C for 72 hr, warmed over 12 hr to −20°C, held at that temperature for 8 hr, then warmed to 4°C for 1 hr. The fixative was removed and the samples rinsed 4 x with cold acetone, following which they were infiltrated with Epon-Araldite resin (Electron Microscopy Sciences, Port Washington PA) over 48 hr. BM was flat-embedded between two Teflon-coated glass microscope slides and the resin polymerized at 60°C for 48 hr.

## TEM, Dual-Axis Tomography and analysis

Flat-embedded BM was observed with a stereo dissecting microscope to ascertain and select well-preserved samples. Suitable tissues were extracted with a scalpel and glued to the tips of plastic sectioning stubs. Semi-thick (400 nm) serial sections were cut with a UC6 ultramicrotome (Leica Microsystems) using a diamond knife (Diatome, Ltd. Switzerland). Sections were placed on Formvar-coated copper-rhodium slot grids (Electron Microscopy Sciences) and stained with 3% uranyl acetate and lead citrate. Gold beads (10 nm) were placed on both surfaces of the grid to serve as fiducial markers for subsequent image alignment. Grids were placed in a dual-axis tomography holder (Model 2040, E.A. Fischione Instruments, Export, PA) and imaged with a Tecnai TF30ST-FEG transmission electron microscope (300 KeV; ThermoFisher Scientific) equipped with a 2k × 2 k CCD

camera (XP1000; Gatan, Inc, Pleasanton CA). Tomographic tilt-series and large-area montaged overviews were acquired using the SerialEM software package (*Mastronarde, 2005*). For tomography, samples were tilted + /- 64° and images collected at 1° intervals. The grid was then rotated 90° and a similar series taken about the orthogonal axis. Tomographic data were calculated, analyzed and modeled using the IMOD software package (*Kremer et al., 1996*; *Mastronarde, 2008*) on MacPro computers (Apple, Inc, Cupertino, CA).

## Acknowledgements

We thank Andres Collazo at the Caltech Biological Imaging Facility for use of confocal and light sheet microscopes and help with image capture and analysis, Carol Garland and the Caltech Kavli Nanoscience Institute for aid in maintaining the TF30 electron microscope, and the Gordon and Betty Moore and Beckman Foundations for gifts to Caltech to support electron microscopy. This research was supported by the Rosalind W Alcott post-doctoral fellowship (Caltech) and startup funding from the University of Illinois at Urbana-Champaign School of Molecular and Cellular Biology (CK), NIAID 1R01AI100652-01A1, the UCLA AIDS Institute, and the UCLA Center for AIDS Research NIH/NIAID AI028697 (DSA), and the National Institute of General Medical Sciences (2 P50 GM082545-08) and California HIV/AIDS Research Program (ID15-CT-017) (PJB). The funders had no role in study design, data collection and interpretation, or the decision to submit the work for publication. The opinions, findings, and conclusions herein are those of the authors and do not necessarily represent those of The Regents of the University of California, or any of its programs.

## Additional information

### Competing interests

Pamela J Bjorkman: Reviewing editor, *eLife*. The other authors declare that no competing interests exist.

### Funding

| Funder | Grant reference number | Author |
|---|---|---|
| National Institute of Allergy and Infectious Diseases | 1R01AI100652-01A1 | Dong Sung An |
| National Institute of Allergy and Infectious Diseases | AI028697 | Dong Sung An |
| National Institute of General Medical Sciences | 2 P50 GM082545-08 | Pamela J Bjorkman |
| California HIV/AIDS Research Program | ID15-CT-017 | Pamela J Bjorkman |
| California Institute of Technology | Rosalind W. Alcott Postdoctoral Fellowship | Collin Kieffer |
| University of Illinois at Urbana-Champaign | School of Molecular and Cellular Biology Startup Funding | Collin Kieffer |

The funders had no role in study design, data collection and interpretation, or the decision to submit the work for publication.

### Author contributions

Mark S Ladinsky, Conceptualization, Formal analysis, Investigation, Visualization, Methodology, Writing—original draft, Writing—review and editing; Wannisa Khamaikawin, Conceptualization, Formal analysis, Investigation, Visualization, Methodology, Writing—review and editing; Yujin Jung, Samantha Lin, Jennifer Lam, Investigation, Methodology; Dong Sung An, Conceptualization, Formal analysis, Funding acquisition, Visualization, Writing—review and editing; Pamela J Bjorkman, Collin

Kieffer, Conceptualization, Funding acquisition, Investigation, Visualization, Methodology, Writing—original draft, Writing—review and editing

### Author ORCIDs
Pamela J Bjorkman [iD] http://orcid.org/0000-0002-2277-3990
Collin Kieffer [iD] https://orcid.org/0000-0001-9051-3819

### Ethics
Animal experimentation: Animals were maintained at the UCLA CFAR Humanized Mouse Core laboratory in accordance with a protocol approved by the UCLA Animal Research Committee. Experiments conformed to all relevant regulatory standards.(UCLA ARC # 2007-092-41A).

### Decision letter and Author response
Decision letter https://doi.org/10.7554/eLife.46916.019
Author response https://doi.org/10.7554/eLife.46916.020

## Additional files

### Supplementary files
• Transparent reporting form DOI: https://doi.org/10.7554/eLife.46916.017

### Data availability
Source data files have been provided for graphs from Figure 1.

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
