## [Decision Letter]

**Acceptance summary:**

The study takes advantage of advances in imaging to study HIV dissemination in the bone marrow in a humanized mouse model of HIV infection. They combine state of the art light and electron microscopy techniques to characterize 3 mechanisms of virus spread among human bone marrow cells the BLT mice: (i) virus budding from T cell and macrophage membranes, (ii) virus producing T-cells directly contacting target cells, and (iii) macrophages engulfing HIV-1-producing cells and virus budding within intracellular compartments that fused to invaginations with access to the extracellular space. Their evidence for cell-cell spread along a viral synapse supports previous models based on viral spread in cell culture models. Thus, this study addresses the important topic of cell-cell HIV spread, which is an area that lacks clarity in HIV transmission biology outside of the context of cell culture models.

**Decision letter after peer review:**

[Editors’ note: this article was originally rejected after discussions between the reviewers, but the authors were invited to resubmit after an appeal against the decision.]

Thank you for submitting your work entitled "Mechanisms of virus dissemination in bone marrow of HIV-1-infected humanized BLT mice" for consideration by *eLife*. We apologize for the delay in examining your manuscript. Your article has been reviewed by three peer reviewers, one of whom is a member of our Board of Reviewing Editors, and the evaluation has been overseen by Senior Editor. The reviewers have opted to remain anonymous.

Our decision has been reached after consultation between the reviewers. Based on these discussions and the individual reviews below, we regret to inform you that your work will not be considered further for publication in *eLife*.

The reviewers agreed that the images are interesting and that this is a unique approach and model. But there was also agreement that the nature of the experiments and the lack of controls made it hard to draw conclusions. In particular, a robust set of experiments with uninfected controls and reagent controls were viewed as important in drawing conclusions and making sure these results were not artefacts. So while there was enthusiasm for the approach and questions, the data seemed a bit premature for publication in *eLife*.

Reviewer #1:

The study of Ladinsky and colleagues takes advantage of advances in imaging to study HIV dissemination in the bone marrow in a humanized mouse model of HIV infection. HIV positive cells in bone marrow could be detected within the first few days of infection in this model. These studies suggest that virions originate from T cells and macrophages and these budding virions were often in proximity to the ESCRT machinery consistent with the role for these pathways in HIV egress. They further showed evidence for cell-cell spread along a viral synapse, which has been proposed as a key mechanism of viral spread in cell culture models. They further showed a novel mechanism of virus budding within macrophages. Macrophages were a major infected cell type in bone marrow in this model and they participated in uptake of infected cells.

Overall, these studies advance some of what we know from cell culture to the more complex setting of the humanized mouse, where the role of interactions between cells in HIV spread can be more readily examined. While it is unclear whether this model predicts the events that occur in HIV infected humans, the same can certainly be said for cell culture models and for NHP studies and this model has distinct advantages – both in studying infected human cells in human tissue and in the ability to visualize events. The findings of evidence of cell-cell spread adds to an important debate in the field as to whether cell-free of cell-cell spread is most important in HIV, although this study does not go particularly far in this aspect.

I am not well positioned to comment on the technical aspects and will leave that to the other reviewers. This does seem a nice complement to NHP studies where the viruses that are used are highly adapted and often unusual.

The main recommendation I have is a larger discussion that brings in more of what is known in humans, if anything, on these cell populations in BM, cell-free versus cell spread etc and also how this compares to data from NHPs. Likewise, what is known about macrophage as the source of virus spread in humans? As written, the discussion feels somewhat narrow and repetitive with the results. Of course, the paper would be stronger if they could actually show macrophages are driving spread from BM to tissue.

Reviewer #2:

The submitted manuscript "Mechanisms of virus dissemination in bone marrow of HIV-1-infected humanized BLT mice" describes the results of an elegant set of experiments that combine state of the art light and electron microscopy techniques to characterize 3 mechanisms of virus spread among human bone marrow cells the BLT mice. This study builds on the authors' previous work studying HIV in the gut and lymphoid tissues of these mice. The results reported are novel in that 3 routes of virus spread were found including (i) virus budding from T-cell and macrophage membranes, (ii) virus producing T-cell uropods directly contacting target cells, and (iii) macrophages engulfing HIV-1-producing cells and virus budding within intracellular compartments that fused to invaginations with access to the extracellular space. This is the first time that all 3 methods of virus spread have been documented in-vivo, although a concern is that there are limits of interpreting static images to define dynamic processes. There are also concerns about absence of controls, both uninfected animals and reagent controls.

Specific comments.

1) "Virus" versus "virion" (or "viral particle"). The terms virus and virion can often be used interchangeably but not when describing ultrastructural findings. It would clarify the findings if the authors used the term "virion" when describing viral particles visualized by EM.

2) The observation that electron dense bands consistent with ESCRT bands is interesting but probably does not warrant as much space in the discussion unless immuno-EM is used to confirm that ECRT proteins form the bands.

3) There is little discussion of the CD4-/CD68- p24+ cell population even though they make up considerable percentage of the infected cells. Do the authors have an explanation for what type cells these are? Of note, it was recently shown that BM myeloid derived suppressor cells can be infected with SIV in vivo.

4) What negative controls were used to ensure that the Mab staining was specific? Was an irrelevant anybody used, primary antibody omitted, negative tissues tested…?

Reviewer #3:

This manuscript by Ladinsky et al. examines the with electron microscopy and light sheet fluorescence microscopy the presence of signs of HIV infection in humanized mouse bone marrow. The application of tissue clearing approaches and electron tomography within this compartment and using the BLT model is novel and presents interesting images of what is possible within this system. There are however, significant limitations of the study which should be addressed. Often, the authors overinterpret aspects of the data and state their imprecations as fact. There is also limited comparisons with non-infected or human BM comparators to allow one to say what might be specific to HIV infected BM. I discuss these limitations further below.

A limitation of the data as it presented here is how to place this into context with a human bone marrow. Something that would help would be to try to place the data into context with respect to how well this compartment serves as a model for human bone marrow. Perhaps some comparative data with conventional pathology images, bone marrow smears, flow cytometry, would help to define the system in comparison with normal and/or HIV infected human BM. It would be nice to assess the overall architecture already with regards to cellular density and the representation of the different cell types.

Many of the images are certainly intriguing, and of high quality, however, the conclusions made from them infer activity that one cannot establish with fixed images. Some of the nomenclature and descriptions and cell calling methods appear to use a best guess approach.

1) RE: Dissemination mechanism #1. It is not fair to say synchronous as this implies the speed with which the viral release has occurred. Figure 3B, authors speculate that it is interesting that the HIV bud is so close to the nucleus, but it could be that when the bud formed it was not in close proximity to the nucleus. The morphology of HIV infected cells is highly dynamic, so it may not be so surprising that a bud is close to or far away from any cellular structure.

2) RE: Dissemination mechanism 2. RE: Virions attached to uropods. While it is an interesting hypothesis that the structures are uropods, uropods are defined largely by the morphology of migrating cell and are difficult to define in a static image without surface molecules. It may be a leap to call these cellular extensions uropods. Figure 4F, It is unclear how the authors define the features of a synapse. How does one define synapse in their approach?

3) RE: Dissemination mechanism 3. Macrophages phagocytosis. It would be nice to characterize the frequency with which one sees phagocytosis of infected cells, versus non-infected cells. How unique or specific is this to HIV infected scenarios? Figure 5C the cell is referred to as a polymorphonuclear macrophage, could this be a neutrophil with few granules?

There appear to be large spaces between the cells which may be artifact of the fixation process which may affect conclusions about what cells are physically interacting or not. The authors should comment on how reflective the spaces are of natural state of the BM and how this may impact their conclusions about cell-cell interactions.

Overall, while the imaging modalities are innovative and of high quality in general, the novelty of the observations is modest. Some of the conclusions made and discussion infer temporal relationships which are not well supported, and the naming process seems a little arbitrary, how does one truly distinguish the different cell types. Additional comparison with what is normal in both the uninfected BLT mouse as well as in human BM would help to put the observations in a greater context.

---

## [Author Response]

[Editors’ note: the author responses to the first round of peer review follow.]

The reviewers agreed that the images are interesting and that this is a unique approach and model. But there was also agreement that the nature of the experiments and the lack of controls made it hard to draw conclusions. In particular, a robust set of experiments with uninfected controls and reagent controls were viewed as important in drawing conclusions and making sure these results were not artefacts. So while there was enthusiasm for the approach and questions, the data seemed a bit premature for publication in eLife.Reviewer #1:Overall, these studies advance some of what we know from cell culture to the more complex setting of the humanized mouse, where the role of interactions between cells in HIV spread can be more readily examined. While it is unclear whether this model predicts the events that occur in HIV infected humans, the same can certainly be said for cell culture models and for NHP studies and this model has distinct advantages – both in studying infected human cells in human tissue and in the ability to visualize events. The findings of evidence of cell-cell spread adds to an important debate in the field as to whether cell-free of cell-cell spread is most important in HIV, although this study does not go particularly far in this aspect.

We thank the reviewer for their appreciation of our studies. We wish to highlight that we were able to detect examples of both cell-free and cell-associated virions by EM, supporting the idea that both forms of HIV-1 transmission do occur in tissue. We also wish to point out that the reason we didn’t study cell-cell spread extensively is that it is rare to find examples of cell-cell spread using EM in BM and other tissues. This is discussed for BM in the submitted paper and summarized for other tissues in the Introduction of the revised paper, citing our previous papers (Ladinsky et al., 2014 and Kieffer et al., 2017) and comparing with our EM studies of murine leukemia virus, in which case we found numerous examples of cell-to-cell transmission (Sewald et al., 2015).

I am not well positioned to comment on the technical aspects and will leave that to the other reviewers. This does seem a nice complement to NHP studies where the viruses that are used are highly adapted and often unusual.

We agree that one advantage of the hu-mouse system for studies of infection is that we can infect with HIV-1 rather than a SHIV, as required for NHP studies.

The main recommendation I have is a larger discussion that brings in more of what is known in humans, if anything, on these cell populations in BM, cell-free versus cell spread etc and also how this compares to data from NHPs. Likewise, what is known about macrophage as the source of virus spread in humans? As written, the discussion feels somewhat narrow and repetitive with the results. Of course, the paper would be stronger if they could actually show macrophages are driving spread from BM to tissue.

This is a good suggestion. The revised Discussion includes a summary of what is known about these cell populations in human and NHP BM and discusses cell-free versus cell-to-cell spread of HIV. Unfortunately, we cannot think of an experimental technique that would definitively show that macrophages are driving spread from BM to tissue. We believe that experiments to address this question would likely involve generation of a novel mouse model, which would require months, if not years, and is outside of our field of expertise. Given that we cannot prove that macrophages drive the spread of HIV from BM to tissue, we have revised the paper to emphasize that this is a speculation.

Reviewer #2:The submitted manuscript "Mechanisms of virus dissemination in bone marrow of HIV-1-infected humanized BLT mice" describes the results of an elegant set of experiments that combine state of the art light and electron microscopy techniques to characterize 3 mechanisms of virus spread among human bone marrow cells the BLT mice. This study builds on the authors' previous work studying HIV in the gut and lymphoid tissues of these mice. The results reported are novel in that 3 routes of virus spread were found including (i) virus budding from T-cell and macrophage membranes, (ii) virus producing T-cell uropods directly contacting target cells, and (iii) macrophages engulfing HIV-1-producing cells and virus budding within intracellular compartments that fused to invaginations with access to the extracellular space. This is the first time that all 3 methods of virus spread have been documented in-vivo, although a concern is that there are limits of interpreting static images to define dynamic processes. There are also concerns about absence of controls, both uninfected animals and reagent controls.

We thank the reviewer for their appreciation of our results. In the revised manuscript, we clarified how we can interpret static images to make limited interpretations of dynamic processes such as HIV-1 budding (see also our response to point #1 from Reviewer 3). We discuss uninfected animal controls in response to the final point from Reviewer 3 (see Author response image 5) and reagent controls in response to point 4 from Reviewer 2.

Specific comments.1) "Virus" versus "virion" (or "viral particle"). The terms virus and virion can often be used interchangeably but not when describing ultrastructural findings. It would clarify the findings if the authors used the term "virion" when describing viral particles visualized by EM.

Thank you for this suggestion. We changed the text as suggested.

2) The observation that electron dense bands consistent with ESCRT bands is interesting but probably does not warrant as much space in the discussion unless immuno-EM is used to confirm that ECRT proteins form the bands.

We have used immuno-EM to confirm that these bands include ESCRT proteins. Please see our previous paper (Ladinsky et al., PLoS Path, 2014, https://doi.org/10.1371/journal.ppat.1003899), where we used immuno-EM to show that the CHMP1B and CHMP2A ESCRT-III proteins are present at locations in budding viruses where these bands of electron density form.

3) There is little discussion of the CD4-/CD68- p24+ cell population even though they make up considerable percentage of the infected cells. Do the authors have an explanation for what type cells these are? Of note, it was recently shown that BM myeloid derived suppressor cells can be infected with SIV in vivo.

We added a citation to the recent findings about BM myeloid-derived suppressor cells to the text and discuss that a portion of the CD4-/CD68- p24+ staining could represent these types of cells in addition to other types of cells that would stain similarly.

4) What negative controls were used to ensure that the Mab staining was specific? Was an irrelevant anybody used, primary antibody omitted, negative tissues tested…?

Relevant negative controls were conducted for immunofluorescence experiments, as described in the revised Materials and methods.

Reviewer #3:This manuscript by Ladinsky et al. examines the with electron microscopy and light sheet fluorescence microscopy the presence of signs of HIV infection in humanized mouse bone marrow. The application of tissue clearing approaches and electron tomography within this compartment and using the BLT model is novel and presents interesting images of what is possible within this system. There are however, significant limitations of the study which should be addressed. Often, the authors overinterpret aspects of the data and state their imprecations as fact. There is also limited comparisons with non-infected or human BM comparators to allow one to say what might be specific to HIV infected BM. I discuss these limitations further below.

We thank the reviewer for this assessment and revised the paper to avoid over-interpretations. Below we discuss comparisons to uninfected humanized mouse BM and human BM.

A limitation of the data as it presented here is how to place this into context with a human bone marrow. Something that would help would be to try to place the data into context with respect to how well this compartment serves as a model for human bone marrow. Perhaps some comparative data with conventional pathology images, bone marrow smears, flow cytometry, would help to define the system in comparison with normal and/or HIV infected human BM. It would be nice to assess the overall architecture already with regards to cellular density and the representation of the different cell types.

We refer reviewers to representative conventional pathology images, BM smears, bone marrow aspirations, and flow cytometry of human BM, which can be compared with BM images in our submitted paper and with Author response images 4 and 5. Pathology of normal human BM from a stained clot section (Page 6 in J.M.Gonzalez-Berjon (2010). Bone Marrow and Spleen Examination. *American Registry of Pathology*
http://hemepathreview.com/Heme-Review/Part15-1-BoneMarrowSpleenExamination.pdf). Stained normal human BM smear at 50x magnification (Figure 2 in L.F. Brass (2005). *J. Clin. Invest. 115: 3329-3331*). Aspiration of normal human BM shows cell-type diversity and distribution that are typical of the field and comparable to montaged projection EM images of HIV-infected BLT mouse BM shown in our manuscript and in Author response image 4 (Science Education Resource Center, Carleton College, https://serc.carleton.edu/woburn/overarching/hema_leuk.html). Flow Cytometry of normal human BM (Figure 2 in Ozcivici et al. (2010). *PLoS One 5: e11178*).

Regarding whether we could conduct comparable imaging studies of human BM, we point out that for our electron tomography imaging, intact BM tissue was taken from HIV-infected BLT mice by means of dissection and direct removal from long bones, rather than by needle aspiration. We did this to obtain suitable amounts of tissue and to reduce disruption of cells and their structural organization. Specifically, BM tissues for our EM studies were obtained by removing ~1 cm of the sternum bone immediately after the mouse was sacrificed and then placing it in a fixative to allow biosafe transport from UCLA to Caltech and for biosafety during processing. After preservation with fixative, the sternum was opened and intact BM was removed without disruption of the underlying tissue structure.

However, for histological studies and transplant procedures, BM is typically extracted from human patients by needle aspiration from long bones. To evaluate if we can successfully image human BM extracted by needle aspiration, we processed BM obtained by needle aspiration from living macaques for EM, but found the method unsatisfactory due to overwhelming amounts of contaminating blood from the vasculature introduced into the samples during the procedure as evidenced in Author response image 1 (i.e., we could not find physiologically-relevant regions of BM that were distinct from contaminating mature red blood cells at the EM level). Conversely, we successfully imaged BM from euthanized NHPs when intact BM was prepared from bone samples as described above for the submitted BLT humanized mouse study. Notably, the NHP BM appeared similar to BLT mouse BM, another indication of the physiological relevance of the BLT mouse model for BM studies. Unfortunately, due to the larger size of NHPs compared with mice and discrepancies in viral load and length of infection, we were unable to observe virus particles or infected cells in these samples. We have also been unable to obtain the larger number of NHP bone samples we would require to systematically search for SIV or SHIV infection in NHP BM, likely because the high cost of NHPs for research precludes sacrificing animals to extract the samples we would require for the sort of systematic study we conducted in HIV-infected humanized mice. As mentioned in our manuscript, macrophages were previously found as early and important targets of SIV infection in NHP (Kitagawa et al., 1991). Our current HIV-infected BLT mouse studies support these previous findings and our use of the BLT mouse model to study mechanisms of virus dissemination within BM. Finally, our findings that intact bone samples are required for high resolution EM imaging of both BLT mouse and NHP BM suggests that imaging of human BM would require bone samples from living HIV-infected patients, which we cannot obtain.

**Author response image 1. respfig1:** Comparison of macaque BM extracted by needle aspiration vs. direct removal from a long bone. (**A**) EM overview of a typical field of macaque BM, extracted from femur by needle aspiration. The field is composed entirely of red (RBC) and white (WBC) blood cells in proportions similar to that of whole blood. Regions containing typical BM resident cells were not found. (**B**) EM overview of a typical field of macaque BM, extracted directly from dissected sternum. The field is composed of typical BM resident cells, including osteocytes, stem cells, macrophages (MF) and megakaryocytes (MKC). The field is similar to that of the BLT mouse BM samples shown in our paper.

In theory, it might be possible to conduct comparable imaging studies in human BM if we could obtain bone samples from HIV-infected patients who had a high viral load when they died. We do not know of a source of such samples. Even if we could obtain these samples, we would need to extract the BM within minutes to hours post-mortem and process the samples for EM within a day after extraction for reliable imaging. To illustrate the problem with degradation when samples are not immediately processed, we processed BLT mouse BM after waiting for longer than a day after its original extraction and found that the degradation was too severe for reliable interpretation (see Author response image 2).

**Author response image 2. respfig2:** Comparison of freshly prepared BLT mouse BM vs. similar sample prepared after longer than one day. (**A**) Overview of BM from a sample processed within 1 day of extraction. Cells are well preserved with even contrast and density of cellular components. (**B**) Overview of BM from a sample processed >1 day after extraction. Cellular degradation is evident as disruption and/or distortion of cells, variable contrast, and density of cellular components (particular nuclei).

There are repositories of HIV-infected BM and other tissues at hospitals such as UCLA that might represent a source for EM imaging studies. However, it is unlikely that infected human BM samples in repositories were extracted from bones using the methodology we showed in Figure 2B and 3A is critical for optimal preservation of samples for electron tomography. Although difficult to discern at the histological level, degradation at the cellular level is obvious when observed at the ~7 nm resolution of EM tomography. As such, tissue samples that have gone unprocessed for even 24 hours following collection and initial fixation are typically of questionable quality for EM tomographic studies. In addition, most or all human tissues available from tissue banks have been preserved in ways that render them inappropriate for high-resolution electron tomography imaging. Although not a problem for histological imaging, the fixatives commonly used in tissue repositories (e.g., paraformaldehyde, formalin, and freshfrozen) are known to distort cellular features at the EM ultrastructural level, and such samples would not be suitable for reliable electron tomography studies in which we need to ascertain, e.g., whether an HIV-containing compartment is located entirely within a cell and/or has access to the extracelluar space through narrow channels (as we did in the paper submitted to *eLife* for infected BLT mouse BM). The fact that most or all human tissues, especially HIV-infected tissues, that are available from tissue banks or similar collection and storage facilities have been preserved for specific tests and/or held in aldehyde fixatives for many months or even years renders this source of human tissue impossible for our EM studies.

In summary, it is unlikely that BM from an HIV-infected human subject could be obtained and prepared in a manner that would allow a direct imaging comparison with our BLT mouse studies because (*i*) bone removal and dissection from a living subject is neither ethical nor practical, (*ii*) obtaining suitably “fresh” samples from a deceased infected subject is equally improbable for many of the same reasons as well as logistical ones, and (*iii*) HIV-infected BM samples obtained from tissue repositories are not preserved using protocols that can generate sufficient ultrastructural resolution by electron tomography or tissue clearing and IF. And in all of these cases, there is effectively no likelihood that a human BM sample could be matched with our BLT mouse samples in terms of viral load or stage of infection. For these reasons, we cannot conduct direct EM comparisons of HIV-infected human BM (or SIV/SHIV-infected NHP BM) with HIV-infected BLT mouse BM to address whether the HIV dissemination mechanisms we interpreted from our EM studies of BLT mouse BM occur in other sources of infected BM.

Many of the images are certainly intriguing, and of high quality, however, the conclusions made from them infer activity that one cannot establish with fixed images. Some of the nomenclature and descriptions and cell calling methods appear to use a best guess approach.

A limitation of samples for electron tomography is that when they are preserved in a way to allow the recording of the highest resolution ultrastructural details, it is not possible to use the same sections for immuno-EM. However, as described in the text, many of the cells in BM can be identified by morphology. In addition to the current descriptions of morphology, we added text explaining that the same samples cannot be used for both high resolution tomography and immuno-EM and describing in greater detail the rationale for identification of each cell type based on its morphology.

1) RE: Dissemination mechanism #1. It is not fair to say synchronous as this implies the speed with which the viral release has occurred. Figure 3B, authors speculate that it is interesting that the HIV bud is so close to the nucleus, but it could be that when the bud formed it was not in close proximity to the nucleus. The morphology of HIV infected cells is highly dynamic, so it may not be so surprising that a bud is close to or far away from any cellular structure.

We thank the reviewer for pointing out what is likely mostly a problem with how we described our results. Perhaps our choice of terminology caused confusion. We chose the term “semisynchronous” to denote that the virions are at a similar stage of egress. Our use of the term “synchronized” was intended to imply multiple events happening at the same time. In the case of HIV budding from a T-cell, we are suggesting that multiple buds form within a particular time period (consistent with finding multiple buds at similar states of egress after rapid fixation), and a portion of that time period is defined as when ESCRT bands are present on the necks of the nascent virions. Since the entire budding process may take minutes and the accumulation/function of ESCRT takes a subset of those minutes, we can infer that if all of the buds present on a region of the cell are showing ESCRT bands, they are all more-or-less at the same point in the budding process. Further, the absence of buds with ESCRT bands within that same region of the cell implies that the budding event in toto is initiated and proceeds at multiple points and within roughly the same definable time window. We have re-worded the text in order to make this description more clear and specific.

Regarding the description of the HIV bud near the nucleus, the reviewer correctly points out that the bud may have formed further from the nucleus. We revised the text to address this possibility.

2) RE: Dissemination mechanism 2. RE: Virions attached to uropods. While it is an interesting hypothesis that the structures are uropods, uropods are defined largely by the morphology of migrating cell and are difficult to define in a static image without surface molecules. It may be a leap to call these cellular extensions uropods.

We use the same definition for a uropod as in Sewald et al., Science. 2015 Oct 30; 2015 Oct 1. PMID: 26429886, where it says, “We identified numerous complex membranous protrusions to which many viral particles were localized. These protrusions were often long and formed contacts between both neighbouring and distally located cells. These membrane-rich protrusions originated from a donor cell and were as long as ~10 μm—a structure that was also occasionally observed by intravital imaging. These protrusions were rich in intracellular vesicles and mitochondria, indicating that they represent uropods.”

Figure 4F, It is unclear how the authors define the features of a synapse. How does one define synapse in their approach?

As clarified in the revised text, we define a synapse in our studies as a region of close proximity between a donor cell and a potential target cell, often including zones of direct contact, which facilitates transfer of material from the donor cell only to the target cell.

3) RE: Dissemination mechanism 3. Macrophages phagocytosis. It would be nice to characterize the frequency with which one sees phagocytosis of infected cells, versus non-infected cells. How unique or specific is this to HIV infected scenarios?

Macrophages actively phagocytose many types of cells, including but not limited to cells infected with viruses, oncogenically-transformed cells, and dead/dying cells. In our current BM studies, we can qualitatively say that uninfected cells (i.e., cells with no visible associated virus) outnumbered cells associated with virus by ~20:1. We included this observation in the revised manuscript in subsection “Dissemination mechanism #3: Macrophage phagocytosis of virus-producing T-cells and release of intracellular virions”.

Figure 5C the cell is referred to as a polymorphonuclear macrophage, could this be a neutrophil with few granules?

The reviewer raises a valid point that we previously considered. Neutrophils in BM from BLT hu-mice are indeed polymorphonuclear (see Author response image 4), but they are easily distinguished from macrophages by their smaller size and populations of large, evenly dense granules. Neutrophils from BLT hu-mice were morphologically comparable to human neutrophils (compare Author response image 4 with figure 1 from Brinkmann and Zychlinsky (2012). *J. Cell Biol 198:773-783*). Furthermore, neutrophils lack extensive, pleomorphic surface invaginations that are common to macrophages. Such invaginations appear electron-lucent in BM macrophages (see Figure 2 and Figure 7—figure supplement 1 from our submitted manuscript). For these reasons, we are confident that the cells we identify as macrophages are not neutrophils with few granules.

**Author response image 3. respfig3:** Electron microscopy of neutrophils. Polymorphonuclear neutrophil from HIV-infected BLT mouse BM.

There appear to be large spaces between the cells which may be artifact of the fixation process which may affect conclusions about what cells are physically interacting or not. The authors should comment on how reflective the spaces are of natural state of the BM and how this may impact their conclusions about cell-cell interactions.

Overall cell density in BM is variable by location and noticeably lower than many other lymphoid tissues (https://en.wikipedia.org/wiki/Bone_marrow). Figure 2A from our manuscript can be compared with Author response image 4 that shows an EM overview of BLT mouse BM, histological images from normal healthy bone marrow https://www.shutterstock.com/image-photo/normal-healthy-bone-marrow-depicting-erythroid-74663083, and hematoxylin and eosin histological staining of human BM (Figure 18 B from Travlos, Toxicologic Pathology, 2006; https://doi.org/10.1080/01926230600939856). Note there are spaces between individual cells in all methods of fixation and analysis. Because our tissues for EM were fixed intact (i.e., entire sternums or femurs placed into fixative) and then high pressure frozen, the potential for disruption of BM architecture was minimized as compared to a technique such as needle biopsy to obtain extracted BM. Therefore our fixation protocols should maintain the overall BM architecture equivalently, if not better, than other methods of tissue fixation.

**Author response image 4. respfig4:** EM of BM. EM overview of BLT hu-mouse BM.

In previous papers cited in the submitted paper (Ladinsky et al., 2014 and Kieffer et al., 2017), we found examples of potential cell-to-cell spread of virions in tissues with closely-packed cells (gut, spleen) as well as an abundance of free virus. Because the overall cell density in BM is reduced compared to other lymphoid tissues, we speculate that virus-producing cells contacting other cells were more rare in this tissue, but potentially more directed due to the lower density of total cells in addition to potential target cells.

Overall, while the imaging modalities are innovative and of high quality in general, the novelty of the observations is modest. Some of the conclusions made and discussion infer temporal relationships which are not well supported, and the naming process seems a little arbitrary, how does one truly distinguish the different cell types. Additional comparison with what is normal in both the uninfected BLT mouse as well as in human BM would help to put the observations in a greater context.

We thank the reviewer for their appreciation of the imaging, but respectfully disagree that the temporal relationships we discuss are not well supported and that the novelty of new observations presented in our paper is modest. Regarding temporal relationships, please see our response to point #1. Regarding novelty, we know of no other studies in tissues during early stages of infection that show macrophage ingestion of virus-producing T-cells and presents a plausible mechanism by which virus can then be released from the macrophage. We also know of no other study that conclusively documents macrophages assembling and releasing HIV into completely enclosed intracellular compartments, a topic of interest to HIV researchers as evidenced by previous controversies in the literature on this topic that are described in studies cited in our paper. As described above and in the revised text, we explained how we identified cells based on morphology, further clarified our discussion of temporal events in HIV budding, and discussed our results in the context of published results in human BM. Regarding comparisons with uninfected BM, below we show an EM image of uninfected BLT mouse BM (Author response image 5). We added a sentence to the revised manuscript saying that, with the exception of not including budding or free viruses, uninfected BLT mouse BM is not distinguishable from HIV-infected BLT mouse BM.

**Author response image 5. respfig5:** Montaged overview of a typical field of BM from an uninfected BLT mouse. Cell density and distribution of cell types are indistinguishable from that of similar HIV-infected BM tissue.